# Learning 3D Representations of Molecular Chirality with Invariance to Bond Rotations

**Keir Adams[1], Lagnajit Pattanaik[1], Connor W. Coley[1,2]**
[1]Department of Chemical Engineering
[2]Department of Electrical Engineering and Computer Science
Massachusetts Institute of Technology, Cambridge, MA 02139, USA
`{keir,lagnajit,ccoley}@mit.edu`

## Abstract

Molecular chirality, a form of stereochemistry most often describing relative spatial arrangements of bonded neighbors around tetrahedral carbon centers, influences the set of 3D conformers accessible to the molecule without changing its 2D graph connectivity. Chirality can strongly alter (bio)chemical interactions, particularly protein-drug binding. Most 2D graph neural networks (GNNs) designed for molecular property prediction at best use atomic labels to naïvely treat chirality, while E(3)-invariant 3D GNNs are invariant to chirality altogether. To enable representation learning on molecules with defined stereochemistry, we design an SE(3)-invariant model that processes torsion angles of a 3D molecular conformer. We explicitly model conformational flexibility by integrating a novel type of invariance to rotations about internal molecular bonds into the architecture, mitigating the need for multi-conformer data augmentation. We test our model on four benchmarks: contrastive learning to distinguish conformers of different stereoisomers in a learned latent space, classification of chiral centers as R/S, prediction of how enantiomers rotate circularly polarized light, and ranking enantiomers by their docking scores in an enantiosensitive protein pocket. We compare our model, Chiral InterRoto-Invariant Neural Network (ChIRo), with 2D and 3D GNNs to demonstrate that our model achieves state of the art performance when learning chiral-sensitive functions from molecular structures.

## 1 Introduction

Advances in graph neural networks (GNNs) have revolutionized molecular representation learning for (bio)chemical applications such as high-fidelity property prediction (Huang et al., 2021; Chuang et al., 2020), accelerated conformer generation (Ganea et al., 2021; Mansimov et al., 2019; Simm & Hernandez-Lobato, 2020; Xu et al., 2021; Pattanaik et al., 2020b), and molecular optimization (Elton et al., 2019; Brown et al., 2019). Fueling recent developments have been efforts to model shape-dependent physio-chemical properties by learning directly from molecular conformers (snapshots of 3D molecular structures) or from 4D conformer ensembles, which better capture molecular flexibility. For instance, recent state-of-the-art (SOTA) GNNs feature message updates informed by bond distances, bond angles, and torsion angles of the conformer (Schütt et al., 2017; Klicpera et al., 2020b; Liu et al., 2021; Klicpera et al., 2021). However, few studies have considered the expressivity of GNNs when tasked with learning the nuanced effects of stereochemistry, which describes how the relative arrangement of atoms in space differ for molecules with equivalent graph connectivity.

Tetrahedral (point) chirality is a prevalent form of stereochemistry, and describes the spatial arrangement of chemical substituents around the vertices of a tetrahedron centered on a chiral center, typically a carbon atom with four non-equivalent bonded neighbors. Two molecules differing only in the relative arrangements around their chiral centers are called stereoisomers, or enantiomers if they can be interconverted through reflection across a plane. Enantiomers are distinguished in chemical line drawings by a dashed or bold wedge indicating whether a bond to a chiral center is directed into or out of the page (Figure 1A). Although enantiomers share many properties such as boiling points, electronic energies, and solubility in most solvents, enantiomers can display strikingly different behavior when interacting with external chiral environments. Notably, chirality is critical for

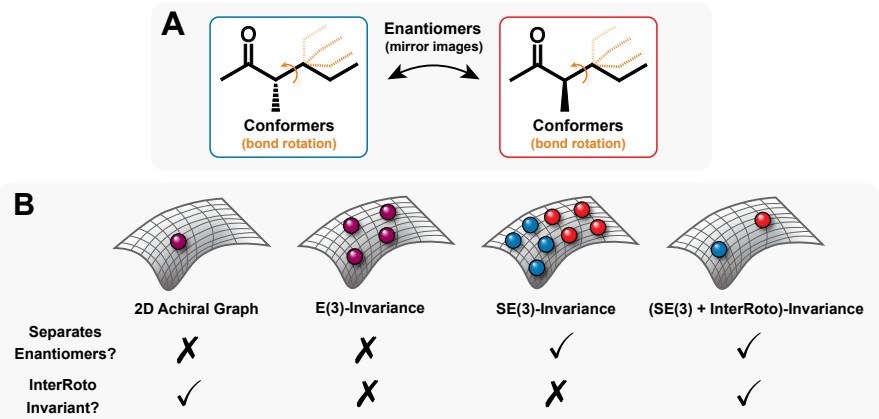

Figure 1: Schematic of how different conformers of two enantiomers (panel A) separate in a learned latent space (panel B) under a 2D (achiral) graph model, E(3)- and SE(3)-invariant 3D models, and ChIRo, which is invariant to rotations of internal molecular bonds (InterRoto-invariance).

drug design (Nguyen et al., 2006; Jamali et al., 1989), where protein-ligand interactions are influenced by ligand chirality, as well as for designing structure-directing agents for zeolite growth (Luis & Beatriz, 2018) and for optimizing asymmetric catalysts (Pfaltz & Drury, 2004; Liao et al., 2018).

Chiral centers are inverted upon reflection through a mirror plane. Consequently, E(3)-invariant 3D GNNs that only consider pairwise atomic distances or bond angles in their message updates, such as SchNet (Schütt et al., 2017) and DimeNet/DimeNet++ (Klicpera et al., 2020a;b), are inherently limited in their ability to distinguish enantiomers (Figure 1B). Although SE(3)-invariant 3D GNNs, such as the recently proposed SphereNet (Liu et al., 2021) and GemNet (Klicpera et al., 2021), can in theory learn chirality, their expressivity in this setting has not been explored.

Alongside the development of 3D GNNs, which process individual 3D conformers, there have been efforts to better represent conformational flexibility by encoding multiple conformers in a 4D ensemble for property prediction (Zankov et al., 2021; 2019; Axelrod & Gomez-Bombarelli, 2021), identifying important conformer poses (Chuang & Keiser, 2020), and predicting solvation energies (Weinreich et al., 2021). Unless in the solid state, molecules are not rigid objects or static 2D graphs, but are flexible structures that rapidly interconvert through rotations of chemical bonds as well as through smaller perturbations such as bond stretching, bending, and wagging. Explicitly modeling this probability distribution over accessible conformer space has the potential to improve the modeling of protein-drug interactions, where the most relevant active pose of the ligand is not known *a priori*, as well as in the prediction of Boltzmann-averages, which depend on a distribution of conformers. One challenge with these methods is selecting which conformers to include in an ensemble: the space of accessible conformations combinatorially explodes with the number of rotatable bonds, and important poses are not known *a priori*. Modeling flexibility with multi-instance methods thus requires explicit conformer enumeration, increasing the cost of training/inference without guaranteeing performance gains. Apart from 2D methods, which ignore 3D information altogether, no studies have explicitly modeled conformational flexibility directly within a model architecture.

To explicitly model tetrahedral chirality and conformational flexibility, we design a neural framework to augment 2D GNNs with processing of the SE(3)-invariant internal coordinates of a conformer, namely bond distances, bond angles, and torsion angles. Our specific contributions are:

- We design a method for graph neural networks to learn the relative orientations of substituents around tetrahedral chiral centers directly from 3D torsion angles
- We introduce a novel invariance to internal rotations of rotatable bonds directly into a model architecture, potentially mitigating the need for 4D ensemble methods or conformer-based data augmentation to treat conformational flexibility of molecules
- We propose a contrastive learning framework to probe the ability of SE(3)-invariant 3D graph neural networks to differentiate stereoisomers in a learned latent space

- Through our ablation study, we demonstrate that a global node aggregation scheme, adapted from Winter et al. (2021), which exploits subgraphs based on internal coordinate connectivity can provide a simple way to improve GNNs for chiral property prediction

We explore multiple tasks to benchmark the ability of our model to learn the effects of chirality. We do not consider common MoleculeNet (Wu et al., 2018) benchmarks, as our focus is on tasks where the effects of chirality are more distinguishable from experimental noise. Our self-supervised contrastive learning task is the first of its kind applied to clustering multiple 3D conformers of different stereoisomers in a latent space. Following Pattanaik et al. (2020a), we also employ a toy R/S labeling task as a necessary but not sufficient test of chiral recognition. For a harder classification task, we follow Mamede et al. (2021) in predicting how enantiomers experimentally rotate circularly polarized light. Lastly, we create a dataset of simulated docking scores to rank small enantiomeric ligands by their binding affinities in a chirality-sensitive protein pocket. We make our datasets for the contrastive learning, R/S classification, and docking tasks available to the public. Comparisons with 2D baselines and the SE(3)-invariant SphereNet demonstrate that our model, **Chiral InterRoto-Invariant Neural Network (ChIRo)**, achieves SOTA in 3D *chiral* molecular representation learning.

## 2 RELATED WORK

**Message passing neural networks.** Gilmer et al. (2017) introduced a framework for using GNNs to embed molecules into a continuous latent space for property prediction. In the typical 2D message passing scheme, a molecule is modeled as a discrete 2D graph $G$ with atoms as nodes and bonds as edges. Nodes and edges are initialized with features $\mathbf{x}_i$ and $\mathbf{e}_{ij}$ to embed initial node states:

$$\mathbf{h}_i^0 = U_0(\mathbf{x}_i, \{\mathbf{e}_{ij}\}_{j \in N(i)}) \tag{1}$$

where $N(i)$ denotes the neighboring atoms of node $i$. In each layer $t$ of message passing, node states are updated with aggregated messages from neighboring nodes. After $T$ layers, graph feature vectors are constructed from some (potentially learnable) aggregation over the learned node states. A readout phase then uses this graph embedding for downstream property prediction:

$$\mathbf{h}_i^{t+1} = U_t(\mathbf{h}_i^t, \mathbf{m}_i^{t+1}), \quad \mathbf{m}_i^{t+1} = \sum_{j \in N(i)} M_t(\mathbf{h}_i^t, \mathbf{h}_j^t, \mathbf{e}_{ij}) \tag{2}$$

$$\hat{y} = \text{Readout}(\mathbf{g}), \quad \mathbf{g} = Agg(\{\mathbf{h}_i^T | i \in G\}) \tag{3}$$

There exist many variations on this basic message passing framework (Duvenaud et al., 2015; Kearnes et al., 2016). In particular, Yang et al. (2019)'s directed message passing neural network (DMPNN, or ChemProp) based on Dai et al. (2016) learns edge-based messages $\mathbf{m}_{ij}^t$ and updates edge embeddings $\mathbf{h}_{ij}^t$ rather than node embeddings. The Graph Attention Network (Veličković et al., 2018) constructs message updates $\mathbf{m}_i^t$ using attention pooling over local node states.

**3D Message Passing and Euclidean Invariances.** 3D GNNs differ in their message passing schemes by using molecular internal coordinates (distances, angles, torsions) to pass geometry-informed messages between nodes. It is important to use 3D information that is at least SE(3)-invariant, as molecular properties are invariant to global rotations or translations of the conformer.

SchNet (Schütt et al., 2017), a well-established network for learning quantum mechanical properties of 3D conformers, updates node states using messages informed by radial basis function expansions of interatomic distances between neighboring nodes. DimeNet (Klicpera et al., 2020b) and its newer variant DimeNet++ (Klicpera et al., 2020a) exploit additional molecular geometry by using spherical Bessel functions to embed angles $\phi_{ijk}$ between the edges formed between nodes $i$ and $j$ and nodes $j$ and $k \neq i$ in the directed message updates to node $i$.

SchNet and DimeNet are E(3)-invariant, as pairwise distances and angles formed between two edges are unchanged upon global rotations, translations, and *reflections*. Since enantiomers are mirror images, SchNet and DimeNet are therefore invariant to this form of chirality. To be SE(3)-invariant, 3D GNNs must consider torsion angles, denoted $\psi_{ixyj}$, between the planes defined by angles $\phi_{ixy}$ and $\phi_{xyj}$, where $i, x, y, j$ are four sequential nodes along a simple path. Torsion angles are negated upon reflection, and thus models considering all torsion angles should be implicitly sensitive to chirality. Flam-Shepherd et al. (2021), Liu et al. (2021) (SphereNet), and Klicpera et al. (2021) (GemNet) introduce 3D GNNs that all embed torsions in their message updates.

Using a complete set of torsion angles provides access to the full geometric information present in the conformer but does not guarantee expressivity when learning chiral-dependent functions. Torsions are negated upon reflection, but any given torsion can also be changed via simple rotations of a rotatable bond–which changes the conformation, but not the molecular identity (i.e., chirality does not change). Reflecting a non-chiral conformer will also negate its torsions, but the reflected conformer can be reverted to its original structure via rotations about internal bonds. To model chirality, neural networks must learn how *coupled* torsions, the set of torsions $\{\psi_{ixyj}\}_{(i,j)}$ that share a bond between nodes $x$ and $y$ (with $x$ or $y$ being chiral), collectively differ between enantiomers.

**E(3)- and SE(3)-Equivariant Neural Networks.** Recent work has introduced *equivariant* layers into graph neural network architectures to explicitly model how global rotations, translations, and (in some cases) reflections of a 3D structure transform tensor properties, such as molecular dipoles or force vectors. SE(3)-equivariant models (Fuchs et al., 2020; Thomas et al., 2018) should be sensitive to chirality, while E(3)-equivariant models (Satorras et al., 2021) will only be sensitive if the output layer is *not* E(3)-invariant. Since we use SE(3)-invariant internal coordinates as our 3D representation, we only compare our model to other SE(3)- or E(3)-invariant 3D GNNs.

**Explicit representations of chirality in machine learning models.** A number of machine learning studies account for chirality through hand-crafted molecular descriptors (Schneider et al., 2018; Golbraikh et al., 2001; Kovatcheva et al., 2007; Valdés-Martiní et al., 2017; Mamede et al., 2021). A naïve but common method for making 2D GNNs sensitive to chirality is through the inclusion of chiral tags as node features. *Local* chiral tags describe the orientation of substituents around chiral centers (CW or CCW) given an ordered list of neighbors. *Global* chiral tags use the Cahn-Ingold-Prelog (CIP) rules for labeling the handedness of chiral centers as R ("rectus") or S ("sinister"). It is unclear whether (and how) models can suitably learn chiral-dependent functions when exposed to these tags as the only indication of chirality. Pattanaik et al. (2020a) propose changing the symmetric message aggregation function in 2D GNNs (sum/max/mean) to an asymmetric function tailored to tetrahedral chiral centers, but this method does not learn chirality from 3D molecular geometries.

**Chirality in 2D Vision and 3D Pose Estimation.** Outside of molecular chirality, there has been work in the deep learning community to develop neural methods that learn chiral representations for 2D image recognition (Lin et al., 2020) and 3D human pose estimation (Yeh et al., 2019). In particular, Yeh et al. (2019) consider integrating equivariance to chiral transforms directly into neural architectures including feed forward layers, LSTMs/GRUs, and convolutional layers.

## 3 CHIRAL INTERROTO-INVARIANT NEURAL NETWORK (CHIRO)

Our model uses a 2D GNN to embed node states based on graph connectivity, two feed-forward networks based on Winter et al. (2021) to encode 3D bond distances and bond angles, a specially designed torsion encoder inspired by Ganea et al. (2021) to explicitly encode chirality with invariance to internal bond rotations (InterRoto-invariance), and a readout phase for property prediction. Figure 2 visualizes how we encode torsion angles to achieve an InterRoto-invariant representation of chirality. The model is visualized in appendix A.4. We implement our network with Pytorch Geometric (Fey & Lenssen, 2019). Our code is available at `https://github.com/keiradams/ChIRo`.

**2D Graph Encoder.** Given a molecular conformer, we initialize a 2D graph $G$ where nodes are atoms initialized with features $\mathbf{x}_i$ and edges are bonds initialized with features $\mathbf{e}_{ij}$. We treat all hydrogens implicitly, use undirected edges, and omit chiral atom tags and bond stereochemistry tags. $\mathbf{x}_i$ include the atomic mass and one-hot encodings of atom type, formal charge, degree, number of hydrogens, and hybridization state. $\mathbf{e}_{ij}$ include one-hot encodings of the bond type, whether the bond is conjugated, and whether the bond is in a ring system. In select experiments, we include global and local chiral atom tags and bond stereochemistry tags for performance comparisons.

Any 2D GNN suffices to embed node states. Following Winter et al. (2021), we use Simonovsky & Komodakis (2017)'s edge convolution (EConv) to embed the node features with the edge features:

$$\mathbf{h}_i^0 = \mathbf{\Theta}\mathbf{x}_i + \sum_{j \in N(i)} \mathbf{x}_j \cdot f_e(\mathbf{e}_{ij}) \tag{4}$$

where $\mathbf{\Theta}$ is a learnable weight matrix and $f_e$ is a multi-layer perceptron (MLP). Node states $\mathbf{h}_i^t \in \mathbb{R}^{h_t}$ are then updated through $T$ sequential Graph Attention Layers (GAT) (Veličković et al., 2018):

$$\mathbf{h}_i^t = \text{GAT}^{(t)}(\mathbf{h}_i^{t-1}, \{\mathbf{h}_j^{t-1}\}_{j \in N(i)}), \;\; t = 1, ..., T \tag{5}$$

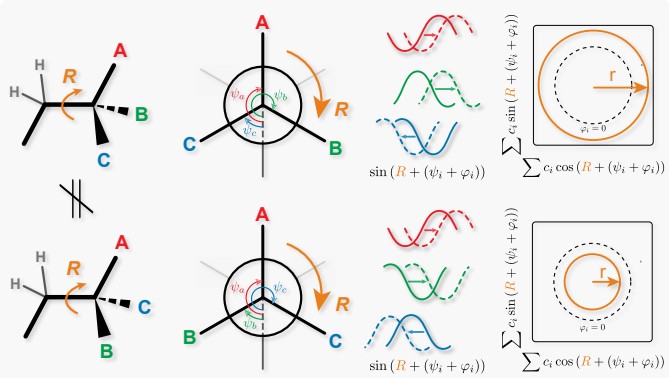

Figure 2: Geometric visualization of how ChIRo learns chiral representations with InterRoto-invariance. Given two enantiomers with three coupled torsions $\psi_a$, $\psi_b$, and $\psi_c$ involving a chiral carbon center, rotating the shared carbon-carbon bond by $R \in [0, 2\pi)$ rotates each torsion together, forming three periodic sinusoids (solid RGB curves). Learning phase shifts $\varphi_i$ for each torsion produces shifted waves (dashed RGB curves) that differ between enantiomers. A weighted sum of the shifted sinusoids results in different degrees of wave interference between enantiomers, yielding different amplitudes of the net waves. These amplitudes, visualized here as the different radii ($r$) of the circles formed by plotting the summed sines against the summed cosines, are invariant to the rotation of the internal carbon-carbon bond. See appendices A.1, A.2 for further details.

**Bond Distance and Bond Angle Encoders.** To embed 3D bond distance and bond angle information, we follow Winter et al. (2021) by individually encoding each bond distance $d_{ij}$ (in Angstroms) and angle $\phi_{ijk}$ into learned latent vectors. We then sum-pool these vectors to get conformer-level latent embeddings $\mathbf{z}_d \in \mathbb{R}^z$ and $\mathbf{z}_\phi \in \mathbb{R}^z$:

$$\mathbf{z}_d = \sum_{(i,j) \in G} f_d(\mathbf{h}_i, \mathbf{h}_j, d_{ij}) + f_d(\mathbf{h}_j, \mathbf{h}_i, d_{ij}) \tag{6}$$

$$\mathbf{z}_\phi = \sum_{(i,j,k) \in G} f_\phi(\mathbf{h}_i, \mathbf{h}_j, \mathbf{h}_k, \cos \phi_{ijk}, \sin \phi_{ijk}) + f_\phi(\mathbf{h}_k, \mathbf{h}_j, \mathbf{h}_i, \cos \phi_{ijk}, \sin \phi_{ijk}) \tag{7}$$

where $f_d$ and $f_\phi$ are MLPs and $(., .)$ denotes concatenation. We maintain invariance to node ordering by encoding both $(i, j)$ and $(j, i)$ for each distance $d_{ij}$, and both $(i, j, k)$ and $(k, j, i)$ for each angle $\phi_{ijk}$. For experiments that mask all internal coordinates (i.e., set all bond lengths and angles to 0), $\mathbf{z}_d$ and $\mathbf{z}_\phi$ represent aggregations of 2D node states based on subgraphs of local atomic connectivity.

**Torsion Encoder.** We specially encode torsion angles to achieve two desired properties: 1) an invariance to rotations about internal molecular bonds, and 2) the ability to learn molecular chirality. Both of these properties depend critically on how the model treats *coupled* torsion angles–torsions that cannot be independently rotated in the molecular conformer. For instance, torsions $\psi_a$, $\psi_b$, and $\psi_c$ in Figure 2 are coupled, as rotating the internal carbon-carbon bond rotates each torsion simultaneously. *Symmetrically* aggregating every redundant torsion does not respect this inherent coupling. However, encoding a set of non-redundant torsions–a minimal set that completely define the structure of the conformer–would make the model sensitive to which arbitrary subset was selected.

Ganea et al. (2021) provide a partial solution to this problem in their molecular conformer generator, which we add to below. Rather than encoding each torsion individually, they compute a weighted sum of the cosines and sines of each coupled torsion (which are redundant), where a neural network predicts the weights[1]. Formally, given a non-terminal (internal) bond between nodes $x$ and $y$ that yields a set of coupled torsions $\{\psi_{ixyj}\}_{(i,j)}$ for $i \in N(x) \setminus \{y\}$ and $j \in N(y) \setminus \{x\}$, they compute:

$$(\alpha_{\cos}^{(xy)}, \alpha_{\sin}^{(xy)}) = \sum_{i \in N(x)\setminus\{y\}} \sum_{j \in N(y)\setminus\{x\}} c_{ij}^{(xy)} \cdot \left( \cos(\psi_{ixyj}), \sin(\psi_{ixyj}) \right) \tag{8}$$

---

[1] Ganea et al. (2021) use an *untrained* network with randomized parameters to predict the weighting coefficients. Instead, we learn the parameters in $f_c$ and compare these approaches in Appendix A.8.

$$c_{ij}^{(xy)} = \sigma\left(f_c(\mathbf{h}_i, \mathbf{h}_x, \mathbf{h}_y, \mathbf{h}_j) + f_c(\mathbf{h}_j, \mathbf{h}_y, \mathbf{h}_x, \mathbf{h}_i)\right) \tag{9}$$

where $f_c$ is an MLP that maps to $\mathbb{R}$. The sigmoid activation $\sigma$, which keeps each $c_{ij}^{(xy)}$ bounded to $(0, 1)$, is our addition. The following formulation is our novel addition to Ganea et al. (2021).

Because rotating the bond $(x, y)$ rotates the torsions $\{\psi_{ixyj}\}$ together, the sinusoids $\{\sin(\psi_{ixyj})\}_{(i,j)}$ and $\{\cos(\psi_{ixyj})\}_{(i,j)}$ have the same frequency. Therefore, the weighted sums of these cosines and sines are also sinusoids, which when plotted against each other as a function of rotating the bond $(x, y)$, form a perfect circle (appendix A.1). Critically, the radius of this circle, $||\alpha_{\cos}^{(xy)}, \alpha_{\sin}^{(xy)}||$, is invariant to the rotation of the bond $(x, y)$ (Figure 2). We therefore encode these radii in order to achieve invariance to how any bond in the conformer is rotated.

The above formulation, as presented, is also invariant to chirality (appendix A.2). To break this symmetry, we add a learned phase shift, $\varphi_{ixyj}$, to each torsion $\psi_{ixyj}$:

$$(\cos\varphi_{ixyj}, \ \sin\varphi_{ixyj}) = \ell_2^{\text{norm}}\left(f_\varphi(\mathbf{h}_i, \mathbf{h}_x, \mathbf{h}_y, \mathbf{h}_j) + f_\varphi(\mathbf{h}_j, \mathbf{h}_y, \mathbf{h}_x, \mathbf{h}_i)\right) \tag{10}$$

$$(\alpha_{\cos}^{(xy)}, \ \alpha_{\sin}^{(xy)}) = \sum_{i \in N(x)\setminus\{y\}} \sum_{j \in N(y)\setminus\{x\}} c_{ij}^{(xy)} \cdot \left(\cos(\psi_{ixyj} + \varphi_{ixyj}), \ \sin(\psi_{ixyj} + \varphi_{ixyj})\right) \tag{11}$$

where $f_\varphi$ is an MLP that maps to $\mathbb{R}^2$ and $\ell_2^{\text{norm}}$ indicates L2-normalization. Because enantiomers have the same 2D graph, the learned coefficients $c_{ij}^{(xy)}$ and phase shifts $\varphi_{ixyj}$ are identical between enantiomers (if $\mathbf{x}_i$ is initialized without chiral tags). However, because enantiomers have different spatial orientations of atoms around chiral centers, the relative values of coupled torsions around bonds involving those chiral centers also differ (Figure 2). As a result, learning phase shifts creates different degrees of wave-interference when summing the weighted cosines and sines for inverted chiral centers. The amplitudes of the net waves (corresponding to radius $||\alpha_{\cos}^{(xy)}, \alpha_{\sin}^{(xy)}||$) will differ between enantiomers, allowing our model to encode different radii for different enantiomers.

With this torsion aggregation scheme, we complete our torsion encoder by individually encoding each internal molecular bond, along with its learned radius, into a latent vector and sum-pooling:

$$\mathbf{z}_\alpha = \sum_{(x,y) \in G} \mathbf{z}_{\alpha_{xy}}, \quad \mathbf{z}_{\alpha_{xy}} = f_\alpha\left(\mathbf{h}_x, \mathbf{h}_y, \left\|\alpha_{\cos}^{(xy)}, \alpha_{\sin}^{(xy)}\right\|\right) + f_\alpha\left(\mathbf{h}_y, \mathbf{h}_x, \left\|\alpha_{\cos}^{(xy)}, \alpha_{\sin}^{(xy)}\right\|\right) \tag{12}$$

**Readout.** For property prediction, we concatenate the sum-pooled node states with the conformer embedding components $\mathbf{z}_d$, $\mathbf{z}_\phi$, and $\mathbf{z}_\alpha$, capturing embeddings of bond lengths, angles, and torsions, respectively. This concatenated representation is then fed through a feed-forward neural network:

$$\hat{y} = f_{\text{out}}\left(\sum_{i \in G} \mathbf{h}_i, \mathbf{z}_d, \mathbf{z}_\phi, \mathbf{z}_\alpha\right) \tag{13}$$

Appendix A.3 explores the option of propagating the learned 3D representations of chirality to node states prior to sum-pooling via additional message passing, treating each $\mathbf{z}_{\alpha_{xy}}$ as a vector of edge-attributes. This stage of chiral message passing (CMP) is designed to help propagate local chiral representations across the molecular graph, and to provide an alternative method of augmenting 2D GNNs which include chiral tags as atom features. However, CMP does not significantly affect ChIRo's performance across the tasks considered and is thus not included in our default model.

## 4 EXPERIMENTS

We evaluate the ability of our model to learn chirality with four distinct tasks: contrastive learning to distinguish between 3D conformers of different stereoisomers in a learned latent space, classifying enantiomer chiral centers as R/S, predicting how enantiomers rotate circularly polarized light, and ranking enantiomers by their docking scores in a chiral protein environment. We compare our model with 2D baselines including DMPNN with chiral atom tags (Yang et al., 2019), and Tetra-DMPNN with and without chiral atom tags (Pattanaik et al., 2020a). We also compare our model to 3D GNN baselines, including the E(3)-invariant SchNet and DimeNet++, and the SE(3)-invariant SphereNet.

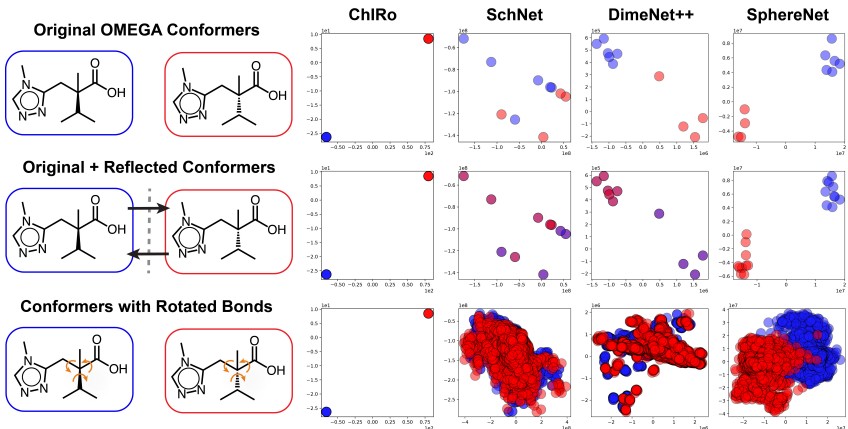

Figure 3: Visualization of how conformers of two enantiomers in the contrastive learning test set cluster in the learned latent space for ChIRo, SchNet, DimeNet++, and SphereNet (top row) using the original OMEGA-generated conformers; (middle row) upon reflecting the conformers, which adds points with inverted chirality (opposite color); and (bottom row) upon rotating internal bonds involving the chiral center. Unlike SchNet, DimeNet++, and even the SE(3)-invariant SphereNet, ChIRo maintains perfect separation between stereoisomers across these conformer transformations.

**Contrastive Learning to Distinguish Stereoisomers.** For contrastive learning, we use a subset of the PubChem3D dataset, which consists of multiple OMEGA-generated conformations of organic molecules with up to 50 heavy atoms and 15 rotatable bonds (Bolton et al., 2011; Hawkins et al., 2010). Our subset consists of 2.98M conformers of 598K stereoisomers of 257K 2D graphs. Each stereoisomer has at least two conformers, and each graph has at least two stereoisomers. We create 70/15/15 training/validation/test splits, keeping conformers corresponding to the same 2D graphs in the same data partition. See appendix A.5 for full training details.

We formulate this task to answer the following question: can the model learn to cluster conformers sharing the same (chiral) molecular identity in a learned latent space, while distinguishing clusters belonging to different stereoisomers of a shared 2D graph? We train each model with a triplet margin loss (Balntas et al., 2016) with a normalized Euclidean distance metric:

$$\mathcal{L}_{\text{triplet}} = \max(0, d(\mathbf{z}_a, \mathbf{z}_p) - d(\mathbf{z}_a, \mathbf{z}_n) + 1), \;\; d(\mathbf{z}_1, \mathbf{z}_2) = \left\| \frac{\mathbf{z}_1}{\|\mathbf{z}_1\|} - \frac{\mathbf{z}_2}{\|\mathbf{z}_2\|} \right\| \tag{14}$$

where $\mathbf{z}_a$, $\mathbf{z}_p$, $\mathbf{z}_n$ are learned latent vectors of anchor, positive, and negative examples. For each triplet, we randomly sample a conformer $a$ of stereoisomer $i$ as the anchor, a conformer $p \neq a$ of stereoisomer $i$ as the positive, and a conformer $n$ of stereoisomer $j \neq i$ as the negative, where $i$ and $j$ share the same 2D graph. For ChIRo, we use $\mathbf{z} = \mathbf{z}_\alpha$. For SchNet, DimeNet++, and SphereNet, we use aggregations of node states (out_channels) as $\mathbf{z}$. We set $\mathbf{z} \in \mathbb{R}^2$ to visualize the latent space.

Figure 3 visualizes how conformers of two enantiomers in the test set separate in the latent space for ChIRo, SchNet, DimeNet++, and SphereNet. We explore how this separation is affected upon reflecting the conformers (which inverts chirality) and rotating bonds around the chiral center. By design, ChIRo maintains perfect separation across these transforms. While SchNet and DimeNet++ may seem to superficially separate the original conformers, the separation collapses upon reflection and rotation due to their E(3)-invariance. SphereNet learns good separation that persists through reflection, but the clusters overlap upon rotation of internal bonds. This emphasizes that 3D GNNs have limited chiral expressivity when not explicitly considering *coupled* torsions in message updates.

**Classifying Tetrahedral Chiral Centers as R/S.** As a toy test of chiral perception for SchNet, DimeNet++, SphereNet and ChIRo, we test whether these 3D models can classify tetrahedral chiral centers as R/S, a simple indication of chirality that can be easily encoded into the initial node features. We use a subset of PubChem3D containing 466K conformers of 78K enantiomers with one tetrahedral chiral center. We create 70/15/15 train/validation/test splits, keeping conformers corresponding to the same 2D graphs in the same partition. We train with a cross-entropy loss and sample one conformer per enantiomer in each batch. See Appendix A.5 for full training details.

Table 1 reports classification accuracies when evaluating on *all* conformers in the test set, without conformer-based averaging. The SE(3)-invariant ChIRo and SphereNet both surpass $98\%$ accuracy, whereas the E(3)-invariant SchNet and DimeNet++ fail to learn this simplest indication of chirality. We emphasize that this classification task, which RDKit trivially solves, is necessary *but not sufficient* to demonstrate that a model can meaningfully learn chiral representations of molecules.

Table 1: R / S classification accuracy on the test set for ChIRo and 3D GNN baselines. Mean and standard deviations are reported across three trials.

| Model | R / S Accuracy (%) ↑ |
|---|---|
| ChIRo | $98.5 \pm 0.2$ |
| SchNet | $54.5 \pm 0.2$ |
| DimeNet++ | $65.7 \pm 2.9$ |
| SphereNet | $98.2 \pm 0.2$ |

**Predicting Enantiomers' Signs of Optical Rotation.** Enantiomers rotate circularly polarized light in different directions, and the sign of rotation designates the enantiomer as $l$ (levorotatory, $-$) or $d$ (dextrorotatory, $+$). Optical activity is an experimental property, and has no correlation with R/S classifications (Table 15, appendix A.7.2). Following Mamede et al. (2021), who used hand-crafted descriptors to predict $l$ / $d$ labels, we extract 30K enantiomers (15K pairs) with their optical activity in a chloroform solvent from the commercial Reaxys database (Reaxys, Accessed Aug. 6-9, 2021), and generate 5 conformers per enantiomer using the ETKDG algorithm in RDKit (Landrum, 2010). Appendix A.7 describes the filtering procedure. We evaluate with 5-fold cross validation, where each pair of enantiomers are randomly assigned to one of the five folds for testing. In each fold, we randomly split the remaining dataset into 87.5/12.5 training/validation splits, assigning enantiomers to the same data partition. We train with a cross-entropy loss. To characterize how conformer data augmentation affects the performance of ChIRo and SphereNet, we employ two training schemes: the first randomly samples a conformer per enantiomer in each batch; the second uses one pre-sampled conformer per enantiomer. In both schemes, we evaluate on all conformers in the test splits. Appendix A.5 specifies full training details.

Table 2 reports the classification accuracies for each model. Notably, ChIRo outperforms SphereNet by a significant $14\%$. Also, whereas ChIRo's performance does not suffer when only one conformer per enantiomer is used, SphereNet's performance drops by $10\%$. This may be due to SphereNet confusing differences in conformational structure versus chiral identities when evaluating pairs of enantiomers (appendix A.10). The consistent performance of ChIRo emphasizes its inherent modeling of conformational flexibility without need for data augmentation. ChIRo also offers significant improvement over DMPNN, demonstrating that our torsion encoder yields a more expressive representation of chirality than simply including chiral tags in node features. The (smaller) improvements over Tetra-DMPNN, which was specially designed to treat tetrahedral chirality in 2D GNNs, suggest that using 3D information to encode chirality is more powerful than 2D representations of chirality.

**Ranking Enantiomers by Binding Affinity in a Chiral Protein Pocket.** *In silico* docking simulations are widely used to screen drug-ligand interactions, especially in high-throughput virtual screens. However, docking scores can be highly sensitive to the conformation of the docked lig-

Table 2: Comparisons of ChIRo with baselines when classifying enantiomers as $l$ / $d$ and ranking enantiomers by their docking scores. Mean accuracies and standard deviations are reported on the test sets across 5 folds ($l$ / $d$) or three trials (enantiomer ranking).

| Model | Accuracy (%) ↑ | |
|---|---|---|
| | $l$ / $d$ | Enantiomer Ranking |
| ChIRo | $\mathbf{79.3 \pm 0.4}$ | $\mathbf{72.0 \pm 0.5}$ |
| ChIRo (1-Conformer) | $79.1 \pm 0.5$ | $71.9 \pm 0.3$ |
| SphereNet | $65.5 \pm 2.4$ | $68.6 \pm 0.3$ |
| SphereNet (1-Conformer) | $55.2 \pm 0.3$ | $63.0 \pm 0.1$ |
| DMPNN with Chiral Tags | $74.4 \pm 0.8$ | $65.9 \pm 0.6$ |
| Tetra-DMPNN (permute) | $70.2 \pm 0.7$ | $66.1 \pm 0.3$ |
| Tetra-DMPNN (concatenate) | $72.6 \pm 0.7$ | $68.5 \pm 0.3$ |
| Tetra-DMPNN (permute) with Chiral Tags | $75.6 \pm 1.3$ | $67.6 \pm 0.6$ |
| Tetra-DMPNN (concatenate) with Chiral Tags | $76.5 \pm 0.8$ | $70.1 \pm 0.5$ |

Table 3: Effects of ablating components of ChIRo on test-set accuracies for the $l / d$ classification and enantiomer docking score ranking tasks. Mean and standard deviations are reported across 5 folds ($l / d$) or 3 repeated trials (enantiomer ranking). The first row indicates the original ChIRo.

| Model Components | | | Accuracy (%) ↑ | |
|---|---|---|---|---|
| Tags | $(d_{ij}, \phi_{ijk}, \psi_{ixyj})$ | $(\mathbf{z}_d, \mathbf{z}_\phi, \mathbf{z}_\alpha)$ | $l / d$ | Enantiomer Ranking |
| X | ✓ | ✓ | $\mathbf{79.3 \pm 0.4}$ | $\mathbf{72.0 \pm 0.5}$ |
| ✓ | ✓ | ✓ | $77.7 \pm 0.8$ | $71.3 \pm 0.7$ |
| ✓ | X | ✓ | $76.2 \pm 0.6$ | $68.4 \pm 0.9$ |
| ✓ | X | X | $70.0 \pm 0.6$ | $63.7 \pm 0.2$ |
| X | X | ✓ | $50.0 \pm 0.0$ | $49.6 \pm 0.3$ |

and, which creates a noisy background from which to extract meaningful differences in docking scores between enantiomers. To partially control for this stochasticity, we source conformers of 200K pairs of small (MW $\leq$ 225, # of rotatable bonds $\leq$ 5) enantiomers with only 1 chiral center from PubChem3D (Bolton et al., 2011). We use AutoDock Vina (Trott & Olson, 2010) to dock each enantiomer in a small docking box (PDB ID: 4JBV) three times, and select pairs for which both enantiomers have a range of docking scores $\leq$ 0.1 kcal/mol across the three trials. Finally, we select pairs of enantiomers whose difference in (best) scores is $\geq$ 0.3 kcal/mol to form a dataset of enantiosensitive docking scores. Each conformer for an enantiomer is assigned the same (best) score. This treats docking score as a stereoisomer-level property. Our final dataset includes 335K conformers of 69K enantiomers (34.5K pairs), which we split into 70/15/15 training/validation/test splits, keeping enantiomers in the same data partition. We train with a mean squared error loss and either randomly sample a conformer per enantiomer in each batch or use one pre-selected conformer per enantiomer. Appendix A.5 specifies full training details. Note that we evaluate the ranking accuracy of the predicted *conformer-averaged* docking scores between enantiomers in the test set.

ChIRo outperforms SphereNet in ranking enantiomer docking scores, achieving an accuracy of 72% (Table 2). This performance is fairly consistent when evaluating ChIRo on subsets of enantiomers which have various differences in their ground-truth scores (Figure 12, appendix A.9). SphereNet once again suffers a drop in performance without conformer data augmentation, whereas ChIRo's performance remains high without such augmentation. ChIRo also outperforms the 2D baselines, with performance gains similar to those in the $l / d$ classification task.

**Ablation Study.** To investigate how the components of ChIRo contribute to its performance on the $l / d$ classification and ranking tasks, we ablate various components including whether stereochemical tags are included in node/edge intialization, whether internal coordinates are masked-out (set to zero), and whether the conformer latent vectors $\mathbf{z}_d$, $\mathbf{z}_\phi$, and $\mathbf{z}_\alpha$ are used in readout (Table 3). Overall, when internal coordinates are masked, including $(\mathbf{z}_d, \mathbf{z}_\phi, \mathbf{z}_\alpha)$ in readout improves performance by $\sim$5% on both tasks. This suggests that if chiral tags are used as the only indication of chirality (i.e., 3D geometry is excluded), using the learned node aggregation scheme provides a simple way to improve the ability of 2D GNNs to learn chiral functions. Notably, adding chiral tags to the unablated ChIRo does *not* improve performance, which is expected given ChIRo's improvements over 2D baselines (Table 2). Omitting chiral tags *and* 3D geometry yields an achiral 2D GNN.

## 5 CONCLUSION

In this work we design a method for improving representations of molecular stereochemistry in graph neural networks through encoding 3D bond distances, angles, and especially torsions. Our torsion encoder learns expressive representations of tetrahedral chirality by processing coupled torsion angles that share a common bond while simultaneously achieving invariance to rotations about internal molecular bonds, diminishing the need for conformer-based data augmentation to capture conformational flexibility. Comparisons to the E(3)-invariant SchNet and DimeNet++ and the SE(3)-invariant SphereNet on a variety of tasks (one self-supervised, three supervised) dependent on molecular chirality demonstrate that ChIRo achieves state-of-the-art in learning chiral representations from 3D molecular structures, while also outperforming 2D GNNs. We leave consideration of other types of stereoisomerism, particularly cis/trans isomerism and atropisomerism, to future work.

## ETHICS STATEMENT

Advancing representations of molecular chirality and conformational flexibility has the potential to accelerate pharmaceutical drug design, renewable energy development, and progress towards energy-efficient and waste-reducing catalysts, among other areas of scientific research. However, in principle, the same advances could also be used for harmful biological, chemical, or materials research and applications thereof.

## REPRODUCIBILITY STATEMENT

Care has been taken to ensure reproducibility of all models and experiments in this paper. In addition to detailing exact model architectures and training details in the appendix, we make source code with experimental setups, model implementations, and random seeds available at `https://github.com/keiradams/ChIRo`. Our GitHub site also contains links to the exact datasets and splits used in each experiment for the contrastive learning, R/S classification, and ranking enantiomers by docking score tasks. Although copyright restrictions prevent us from releasing the dataset and splits for the $l$ / $d$ classification task, we detail our data filtering and processing steps in appendix A.7.

### ACKNOWLEDGMENTS

This research was supported by the Office of Naval Research under grant number N00014-21-1-2195. L.P. thanks the MIT-Takeda fellowship program for financial support. The authors acknowledge the MIT SuperCloud and Lincoln Laboratory Supercomputing Center for providing HPC resources that have contributed to the research results reported within this paper. The authors thank Octavian Ganea and John Bradshaw for providing helpful suggestions regarding the content and organization of this paper.

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

# A  APPENDIX

## CONTENTS

## A.1 WHY DOES PLOTTING $\alpha_{\text{SIN}}^{(xy)}$ VS. $\alpha_{\text{COS}}^{(xy)}$ YIELD A PERFECT CIRCLE?

It is not immediately obvious that plotting $\alpha_{\text{sin}}^{(xy)}$ versus $\alpha_{\text{cos}}^{(xy)}$ forms a perfect circle with a constant radius, as opposed to an ellipse or other shape with non-constant radii. Yet, this result follows from simple trigonometry.

From Equation 11, we have:

$$\alpha_{\text{cos}}^{(xy)} = \sum_{i \in N(x) \setminus \{y\}} \sum_{j \in N(y) \setminus \{x\}} c_{ij}^{(xy)} \cos(\psi_{ixyj} + \varphi_{ixyj})$$

$$\alpha_{\text{sin}}^{(xy)} = \sum_{i \in N(x) \setminus \{y\}} \sum_{j \in N(y) \setminus \{x\}} c_{ij}^{(xy)} \sin(\psi_{ixyj} + \varphi_{ixyj})$$

for a internal molecular bond between nodes $(x, y)$ that forms a set of coupled torsions $\{\psi_{ixyj}\}_{(i,j)}$ where $i \in N(x) \setminus \{y\}$ and $j \in N(y) \setminus \{x\}$. For clarity, we drop the $(xy)$ notation and index each torsion in this set of $n$ coupled torsions by a single index $k$ such that $\{\psi_{ixyj}\}_{(i,j)} = \{\psi_k\}_{k=1}^n$.

We want to show that plotting $\sum_{k=1}^n c_k \sin(R + \psi_k + \varphi_k)$ versus $\sum_{k=1}^n c_k \cos(R + \psi_k + \varphi_k)$, where $R \in [0, 2\pi)$ is a rotation of the bond $(x, y)$, yields a circle with a constant radius invariant to the rotation $R$. As a simple case, consider $n = 2$. We then have:

$$\alpha_{\text{cos}} = c_1 \cos(R + \psi_1 + \varphi_1) + c_2 \cos(R + \psi_2 + \varphi_2) = c_1 \cos(R + \theta_1) + c_2 \cos(R + \theta_2)$$

$$\alpha_{\text{sin}} = c_1 \sin(R + \psi_1 + \varphi_1) + c_2 \sin(R + \psi_2 + \varphi_2) = c_1 \sin(R + \theta_1) + c_2 \sin(R + \theta_2)$$

$\alpha_{\text{cos}}$ can be expanded and simplified as:

$$c_1 \cos(R + \theta_1) + c_2 \cos(R + \theta_2) =$$
$$c_1 \Big( \cos(R) \cos(\theta_1) - \sin(R) \sin(\theta_1) \Big) + c_2 \Big( \cos(R) \cos(\theta_2) - \sin(R) \sin(\theta_2) \Big) =$$
$$a \cos(R) - b \sin(R) = d \cos(R + e)$$

where

$$a = d \cos e = c_1 \cos(\theta_1) + c_2 \cos(\theta_2)$$
$$b = d \sin e = c_1 \sin(\theta_1) + c_2 \sin(\theta_2)$$

Similarly, $\alpha_{\text{sin}}$ can be expanded and simplified as:

$$c_1 \sin(R + \theta_1) + c_2 \sin(R + \theta_2) =$$
$$c_1 \Big( \sin(R) \cos(\theta_1) + \cos(R) \sin(\theta_1) \Big) + c_2 \Big( \sin(R) \cos(\theta_2) + \cos(R) \sin(\theta_2) \Big) =$$
$$a' \sin(R) + b' \cos(R) = d' \sin(R + e')$$

where

$$a' = d' \cos e' = c_1 \cos(\theta_1) + c_2 \cos(\theta_2)$$
$$b' = d' \sin e' = c_1 \sin(\theta_1) + c_2 \sin(\theta_2)$$

Since $d = d'$ and $e = e'$, we have that $\alpha_{\text{cos}} = d \cos(R + e)$ and $\alpha_{\text{sin}} = d \sin(R + e)$. It follows that in the general case ($n \geq 2$):

$$\alpha_{\text{cos}} = \sum_{k=1}^n c_k \cos(\omega t + \psi_k + \varphi_k) = r \cos(R + s)$$

$$\alpha_{\text{sin}} = \sum_{k=1}^n c_k \sin(\omega t + \psi_k + \varphi_k) = r \sin(R + s)$$

for some coefficients $r$ and $s$, where $R \in [0, 2\pi)$ represents some rotation about the internal bond. It is immediately obvious that plotting $\alpha_{\text{sin}}$ versus $\alpha_{\text{cos}}$ for all $R \in [0, 2\pi)$ yields a perfect circle with radius $r$.

## A.2 WHY DO WE NEED PHASE SHIFTS?

At first glance, it is unclear as to why Equations 8-9 are insufficient to learn chirality, e.g. why we need to add phase shifts to distinguish enantiomers. To understand why this is the case, consider an arbitrary stereoisomer containing an internal molecular bond between nodes $x$ and $y$ (with $x$ or $y$ being chiral) that forms a set of $n$ coupled torsions $\{\psi_{ixyj}\}_{(i,j)} = \{\psi_{xy}^{(k)}\}_{k=1}^{n}$ indexed by $k$. Direct computation of the norm $||\alpha_{\cos}^{(xy)}, \alpha_{\sin}^{(xy)}|| = ||\alpha_{\cos}, \alpha_{\sin}||$ yields:

$$||\alpha_{\cos}, \alpha_{\sin}|| = \sqrt{\alpha_{\cos}^2 + \alpha_{\sin}^2}$$

where we have dropped the $(xy)$ notation for clarity. Expanding and squaring Equation 8 gives:

$$\alpha_{\cos}^2 = \sum_{k=1}^{n} c_k^2 \cos^2 \psi_k + \sum_{k=1}^{n} \sum_{l=k+1}^{n} 2 c_k c_l \cos \psi_k \cos \psi_l$$

$$\alpha_{\sin}^2 = \sum_{k=1}^{n} c_k^2 \sin^2 \psi_k + \sum_{k=1}^{n} \sum_{l=k+1}^{n} 2 c_k c_l \sin \psi_k \sin \psi_l$$

Now consider reflecting this stereoisomer across a plane to generate its enantiomer with inverted chiral centers. Reflecting a conformer negates all its torsions, and thus the reflected enantiomer's internal bond between (reflected) nodes $x'$ and $y'$ will form a set of $n$ negated torsions $\{\psi'_{ixyj}\}_{(i,j)} = \{\psi'_{xy}^{(k)}\}_{k=1}^{n}$ where $\psi'_{xy}^{(k)} = -\psi_{xy}^{(k)}$ for each $k$. Because enantiomers have the same 2D graph connectivity, the learned set of coefficients $\{c_k\}$ (which depend only on the permutation invariant node features, see Equation 5) will be identical between enantiomers. It immediately follows from the identities $\sin(-x) = -\sin(x)$ and $\cos(-x) = \cos(x)$ that $\alpha'_{\cos}^2 = \alpha_{\cos}^2$ and $\alpha'_{\sin}^2 = \alpha_{\sin}^2$. Because $||\alpha_{\cos}, \alpha_{\sin}||$ is invariant to any rotation of the bond $(x, y)$, no matter how we rotate $(x, y)$ or $(x', y')$, $||\alpha_{\cos}, \alpha_{\sin}|| = ||\alpha'_{\cos}, \alpha'_{\sin}||$.

Table 4: Effects of ablating components of ChIRo on test-set accuracies for the $l$ / $d$ classification and enantiomer docking score ranking tasks, when chiral message passing is included prior to readout. Mean and standard deviations are reported across 5 folds ($l$ / $d$) or 3 repeated trials (enantiomer ranking). The first row indicates the original ChIRo without CMP.

| Model Components | | | | Accuracy (%) ↑ | |
|---|---|---|---|---|---|
| CMP | Tags | $(d_{ij}, \phi_{ijk}, \psi_{ixyj})$ | $(\mathbf{z}_d, \mathbf{z}_\phi, \mathbf{z}_\alpha)$ | $l$ / $d$ | Enantiomer Ranking |
| X | X | ✓ | ✓ | $\mathbf{79.3 \pm 0.4}$ | $\mathbf{72.0 \pm 0.5}$ |
| ✓ | X | ✓ | ✓ | $78.5 \pm 0.5$ | $71.5 \pm 0.5$ |
| ✓ | ✓ | ✓ | ✓ | $77.0 \pm 0.5$ | $71.7 \pm 0.6$ |
| ✓ | ✓ | X | ✓ | $75.1 \pm 0.3$ | $69.8 \pm 0.6$ |
| ✓ | ✓ | X | X | $75.0 \pm 0.9$ | $68.9 \pm 0.4$ |
| ✓ | X | X | ✓ | $50.0 \pm 0.0$ | $49.9 \pm 0.3$ |

## A.3 Chiral Message Passing

Tetrahedral chirality is fundamentally a node-level property. Yet, we have treated chirality through the lens of torsions and internal bonds involving chiral centers. To propagate the learned representations of chirality to node states, we may perform an (optional) additional phase of message passing, treating each $\mathbf{z}_{\alpha_{xy}}$ as a vector of edge-attributes. This might allow the model to propagate chiral information across the graph, i.e., help learn the global effects of local chirality. This phase may be included prior to readout, and uses a single EConv layer followed by a sequence of $T_{\text{CMP}}$ GAT layers:

$$\mathbf{h}_i^{\text{CMP}} = \text{GAT}_{\text{CMP}}^{T_{\text{CMP}}} \circ ... \circ \text{GAT}_{\text{CMP}}^1 \left( \boldsymbol{\Theta}_{\text{CMP}} \, \mathbf{h}_i^T + \sum_{j \in N(i)} \mathbf{h}_j^T \cdot f_{\text{CMP}}(\mathbf{z}_{\alpha_{ij}}) \right) \qquad (15)$$

During readout, these updated node states $\mathbf{h}_i^{\text{CMP}}$ are summed rather than the (non-chiral) node states.

Table 4 reports the effects of including chiral message passing on ChIRo's performance on the $l$ / $d$ classification task and the ranking enantiomers by docking scores task, along with the effects of ablating other components of ChIRo. Although CMP does not improve performance (and marginally hurts) the unablated ChIRo, using CMP in place of $(\mathbf{z}_d, \mathbf{z}_\phi, \mathbf{z}_\alpha)$ during readout provides another option of augmenting 2D GNNs that solely use chiral tags as the only indication of chirality.

## A.4 Model Architectures

### A.4.1 ChIRo

Figure 4 visualizes the full architecture of ChIRo. Table 5 specifies the hyperparameters chosen for each task. See appendix A.6 for details on hyperparameter optimizations.

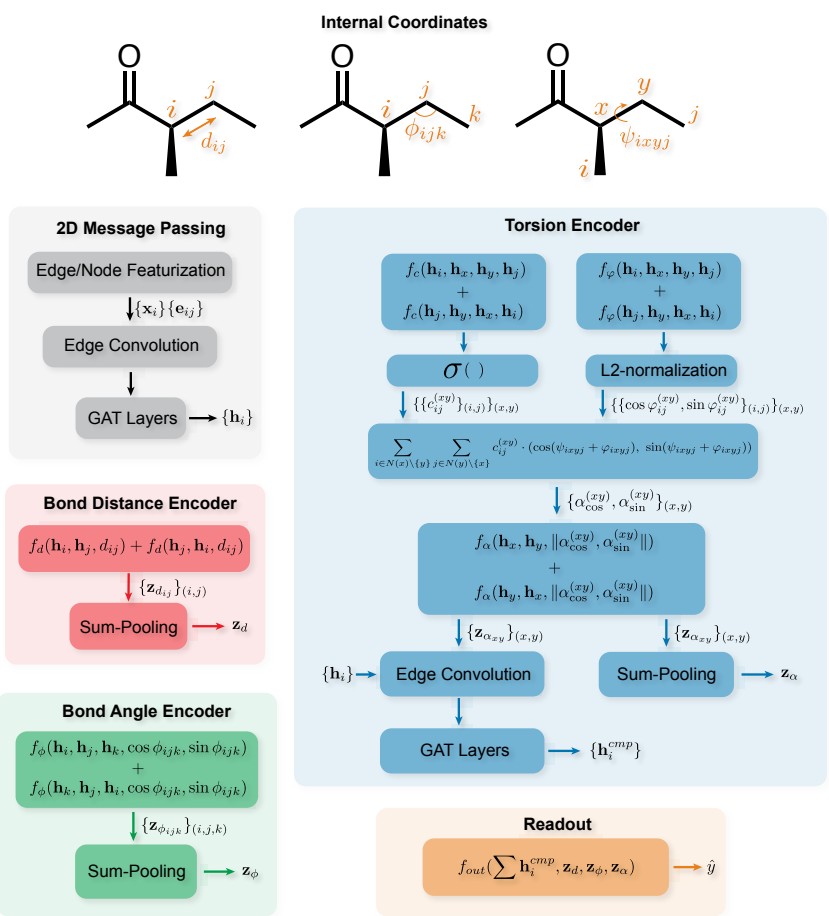

Figure 4: Architecture of ChIRo

Table 5: Hyperparameters optimized for ChIRo on each task. Parameters for chiral message passing are shown, although chiral message passing is omitted in the default version of ChIRo.

| Hyperparameters | Task | | | |
|---|---|---|---|---|
| | Contrastive | R / S | $l / d$ | Ranking |
| Node Features Dimension | | 52 | | |
| Edge Features Dimension | | 14 | | |
| All MLP Hidden Activations | | LeakyReLU | | |
| All MLP Output Activations | | Identity | | |
| EConv MLP Hidden Layer Size | 32 | 64 | 64 | 32 |
| EConv MLP # Hidden Layers | 2 | 1 | 1 | 2 |
| $\mathbf{h}_0, \mathbf{h}_T, \mathbf{h}_t^{\text{CMP}}$ Dimension | 64 | 32 | 32 | 64 |
| $\mathbf{h}_t, t = 1, ..., T - 1$ Dimension | 64 | 64 | 32 | 64 |
| # GAT Layers | 2 | 3 | 2 | 2 |
| # GAT Heads | 4 | 4 | 2 | 4 |
| $f_d$ Hidden Layer Size | – | 128 | 32 | 64 |
| $f_d$ # Hidden Layers | – | 2 | 2 | 2 |
| $f_\phi$ Hidden Layer Size | – | 128 | 32 | 64 |
| $f_\phi$ # Hidden Layers | – | 2 | 2 | 2 |
| $f_\alpha$ Hidden Layer Size | 64 | 128 | 32 | 64 |
| $f_\alpha$ # Hidden Layers | 2 | 2 | 2 | 2 |
| $f_c$ Hidden Layer Size | 64 | 128 | 32 | 64 |
| $f_c$ # Hidden Layers | 2 | 2 | 2 | 2 |
| $f_\varphi$ Hidden Layer Size | 64 | 256 | 256 | 256 |
| $f_\varphi$ # Hidden Layers | 2 | 2 | 3 | 2 |
| $f_{\text{out}}$ Hidden Layer Size | – | 64 | 64 | 128 |
| $f_{\text{out}}$ # Hidden Layers | – | 2 | 2 | 2 |
| $z_d, z_\phi, z_\alpha$ Dimension | 2 | 64 | 8 | 8 |
| CMP EConv Hidden Layer Size | – | 32 | 256 | 256 |
| CMP EConv # Hidden Layers | – | 1 | 3 | 2 |
| # CMP GAT Layers | – | 3 | 3 | 3 |
| # CMP GAT Heads | – | 2 | 2 | 2 |
| $\gamma_{\text{aux}}$ | 8.25e-4 | 6.86e-3 | 1.86e-3 | 8.25e-4 |
| Learning Rate | 6.06e-4 | 5.69e-4 | 1.28e-4 | 6.06e-4 |
| Batch Size | 32 | 16 | 16 | 32 |
| # Epochs | 50 | 100 | 100 | 150 |

Table 6: Hyperparameters selected for SphereNet on each task. See appendix A.6 for hyperparameter optimizations.

| Hyperparameters | Task | | | |
|---|---|---|---|---|
| | Contrastive | R / S | $l$ / $d$ | Ranking |
| Readout MLP Hidden Activations | | LeakyReLU | | |
| Readout MLP Output Activation | | Identity | | |
| Readout MLP Hidden Size | – | 256 | 64 | 64 |
| Readout MLP # of Hidden Layers | – | 4 | 2 | 2 |
| hidden_channels | 128 | 256 | 64 | 256 |
| out_channels | 2 | 64 | 64 | 32 |
| cutoff | 5.0 | 5.0 | 5.0 | 5.0 |
| num_layers | 4 | 4 | 4 | 5 |
| int_emb_size | 64 | 32 | 128 | 64 |
| basis_emb_size_dist | 8 | 8 | 8 | 8 |
| basis_emb_size_angle | 8 | 8 | 8 | 8 |
| basis_emb_size_torsion | 8 | 8 | 8 | 8 |
| out_emb_channels | 256 | 128 | 64 | 32 |
| num_spherical | 7 | 7 | 7 | 7 |
| num_radial | 6 | 6 | 6 | 6 |
| envelope_exponent | 5 | 5 | 5 | 5 |
| num_before_skip | 1 | 1 | 1 | 1 |
| num_after_skip | 2 | 2 | 2 | 2 |
| num_output_layers | 3 | 3 | 3 | 3 |
| Learning Rate | 1e-4 | 1.54e-4 | 4.79e-4 | 1.40e-4 |
| Batch Size | 32 | 64 | 16 | 32 |

### A.4.2 3D BASELINES

To adapt SchNet, DimeNet++, and SphereNet for our tasks, we increase the dimensionality of their respective aggregated node embeddings ("out_channels") and use this aggregation as a latent vector either for direct use in Equation 14 (for self-supervised contrastive learning), or for input to a feed-forward readout MLP for downstream regression/classification. For all three 3D GNNs, we use their default node featurizations, using the atomic number as the only node feature.

Table 6 lists the hyperparameters used for SphereNet on all four tasks. Tables 7 and 8 list the hyperparameters used for SchNet and DimeNet++ on the contrastive learning and R / S classification tasks.

Table 7: Hyperparameters selected for SchNet on each task. Apart from the number of out channels and the inclusion of a readout MLP, the default SchNet architecture was used. Note that we used the same readout MLP architecture as in the optimized SphereNet.

| Hyperparameters | Task | |
| --- | --- | --- |
| | Contrastive | R / S |
| Readout MLP Hidden Activations | – | LeakyReLU |
| Readout MLP Output Activation | – | Identity |
| Readout MLP Hidden Size | – | 64 |
| Readout MLP # of Hidden Layers | – | 2 |
| hidden_channels | 128 | 128 |
| out_channels | 2 | 32 |
| num_filters | 128 | 128 |
| num_interactions | 6 | 6 |
| num_gaussians | 50 | 50 |
| cutoff | 10.0 | 10.0 |
| max_num_neighbors | 32 | 32 |
| Learning Rate | 1e-4 | 1e-4 |
| Batch Size | 32 | 32 |

Table 8: Hyperparameters selected for DimeNet++ on each task. Apart from the number of out channels and the inclusion of a readout MLP, the default DimeNet++ architecture was used. Note that we used the same readout MLP architecture as in the optimized SphereNet.

| Hyperparameters | Task | |
| --- | --- | --- |
| | Contrastive | R / S |
| Readout MLP Hidden Activations | – | LeakyReLU |
| Readout MLP Output Activation | – | Identity |
| Readout MLP Hidden Size | – | 64 |
| Readout MLP # of Hidden Layers | – | 2 |
| hidden_channels | 128 | 128 |
| out_channels | 2 | 32 |
| num_blocks | 4 | 4 |
| cutoff | 5.0 | 5.0 |
| int_emb_size | 64 | 64 |
| basis_emb_size | 8 | 8 |
| out_emb_channels | 256 | 256 |
| num_spherical | 7 | 7 |
| num_radial | 6 | 6 |
| envelope_exponent | 5 | 5 |
| num_before_skip | 1 | 1 |
| num_after_skip | 2 | 2 |
| num_output_layers | 3 | 3 |
| Learning Rate | 1e-4 | 1e-4 |
| Batch Size | 32 | 32 |

Table 9: Hyperparameters selected for DMPNN on each task. Following Pattanaik et al., we only optimize hyperparameters for the baseline sum aggregator and extend these hyperparameters to the TetraDMPNN models.

| Hyperparameters | Task | |
|---|---|---|
| | Contrastive | R / S |
| hidden_size | 300 | 300 |
| depth | 3 | 3 |
| dropout | 0.2 | 0.2 |
| Max Learning Rate | 1e-4 | 1e-4 |
| Batch Size | 50 | 50 |

### A.4.3   2D BASELINES

To optimize hyperparameters for all 2D baselines, we directy use the code provided by Pattanaik et al.. We do not make any modifications to the architectures, since we train on similar tasks as the original work.

## A.5 TRAINING DETAILS

This section describes the full training protocols for each task considered in this paper.

### A.5.1 DIMENET++ INITIALIZATIONS

The default parameter initializations for the output blocks of DimeNet++ make DimeNet++ unable to break symmetry between different output channels when the number of output channels is set $> 1$. Thus, we remove the default initializations for the output blocks when training DimeNet++ on the contrastive learning task (out_channels = 2) and the R/S classification task (out_channels = 32).

### A.5.2 SPHERENET DATA PROCESSING ERRORS

When training SphereNet, occasionally the publicly-available implementation of SphereNet failed to process select conformers. Because this occurred for only a tiny fraction of the conformers in the overall datasets, we removed 2D graphs whose stereoisomers contained problematic conformers from the R/S and $l$ / $d$ datasets *for SphereNet only*. We emphasize that this filtering step did not meaningfully change the size of the datasets: only 40 2D graphs (out of 39,256) were removed for the R/S task, and only 6 2D graphs (out of 15,038) were removed for the $l$ / $d$ task. No molecules had to be removed for the ranking enantiomers by docking scores task. For the contrastive learning task, we simply skipped the rare batch during training/inference that contained problematic conformers. Only 452 2D graphs out of 257,743 caused processing errors in the contrastive learning dataset.

### A.5.3 AUXILIARY TORSION LOSS WHEN TRAINING CHIRO

In Equation 10, we predict an angular phase shift $\varphi$ by using $f_\varphi$ to predict $\cos\varphi$ and $\sin\varphi$ separately with a *linear* output activation, and then use $\ell_2$ normalization on the vector $[\cos\varphi, \sin\varphi]$ to ensure that these $\sin$ and $\cos$ (and thus the angle $\varphi$) have correct circular properties, namely that $\varphi \in [0, 2\pi)$. We have chosen to predict angles using $\sin$ and $\cos$ rather than directly predicting a scalar $\varphi \in [0, 2\pi)$ (e.g., through a scaled sigmoid activation) in order to avoid biasing the predicted phase shifts toward 0. However, the $\ell_2$ normalization can also cause the predicted phase shift to be biased toward $0, \pi/2, \pi$, and $3\pi/2$ if one of the *unnormalized* $\cos\varphi$ or $\sin\varphi$ blows up. To prevent this, we follow Jumper et al. (2021) in adding an auxiliary loss during training to encourage the unnormalized $[\cos\varphi, \sin\varphi]$ to fall close to the unit circle:

$$L_{\text{aux}} = \gamma_{\text{aux}}(1 - ||\cos\varphi, \sin\varphi||) \tag{16}$$

where $\gamma_{\text{aux}}$ is a small scalar that is tuned during hyperparameter optimization.

### A.5.4 CONTRASTIVE LEARNING

For contrastive learning, we train with a triplet margin loss with a normalized Euclidean distance metric and a margin of 1 (Equation 14). In each training epoch, we randomly partition the training data into $N/b$ minibatches, where $N$ is the number of distinct *stereoisomers* (not conformers) in the training set and $b$ is the batch size. Before triplet sampling, each minibatch therefore contains $b$ stereoisomers. For each stereoisomer $i = (1, 2, ..., b)$ in the minibatch, we then generate a triplet $(a, p, n)$, where $a$ (anchor) and $p$ (positive) are two randomly selected (without replacement) conformers of stereoisomer $i$, and $n$ (negative) is a randomly selected conformer of stereoisomer $j \neq i$, where $i$ and $j$ share the same 2D graph connectivity. If stereoisomer $i$ has multiple different stereoisomers $j, k, ...$ (all sharing the same 2D graph) present in the training set, one of these stereoisomers is randomly chosen. Each anchor, positive, and negative in each triplet are processed independently by the network to generate $b$ triplets of latent vectors $(\mathbf{z}_a, \mathbf{z}_p, \mathbf{z}_n)$, which are then fed into the triplet margin loss function. Loss contributions from each triplet are averaged to form a loss for the entire batch. In the case of ChIRo, we add the mean auxiliary torsion loss across all *conformers* in the batch to the batch triplet loss.

We use the Adam optimizer (Kingma & Ba, 2014) with a flat learning rate throughout training, but employ gradient clipping with a maximum gradient L2-norm of 10. We train each model for a maximum of 50 epochs, and use the batch-averaged triplet loss (without the auxiliary torsion loss contributions) on the validation set to select the model with the best estimated generalization performance across the 50 epochs. We do not employ dropout or other forms of explicit regularization.

### A.5.5 R / S CLASSIFICATION

For R / S classification, we train each 3D model with a binary cross-entropy loss. In each training epoch, we randomly partition the training data into $N/b$ minibatches, where $N$ is the number of *enantiomers* in the training set and $b$ is the batch size. For each enantiomer in the batch, we randomly sample a conformer for that enantiomer *and* a conformer for its opposite enantiomer with the same 2D graph. This ensures that minibatches contain both enantiomers of the enantiomeric pair such that the stochastic gradient steps consider contributions from both enantiomers. We average the loss contributions for all conformers in the batch. In the case of ChIRo, the mean auxiliary torsion loss across all conformers in the batch is added to the batch cross-entropy loss.

We use the Adam optimizer with a flat learning rate throughout training, but employ gradient clipping with a maximum gradient L2-norm of 10. We train each model for a maximum of 100 epochs, and use the batch-averaged *classification accuracy* on the validation set to select the model with the best estimated generalization performance across the 100 epochs. We do not employ dropout or other forms of explicit regularization.

For testing, we evaluate the classification accuracy on *all* conformers in the test set, and do not use any form of conformer-based averaging or voting.

### A.5.6 *l / d* CLASSIFICATION

For *l / d* classification, we train each model with a binary cross-entropy loss. In each training epoch, we randomly partition the training data into $N/b$ minibatches, where $N$ is the number of *enantiomers* in the training set and $b$ is the batch size. For each enantiomer in the batch, we sample either 1) a random conformer *or* 2) a pre-selected conformer for that enantiomer, *and* either 1) a random conformer *or* 2) a pre-selected conformer conformer for its opposite enantiomer with the same 2D graph. This ensures that minibatches contain both enantiomers of the enantiomeric pair such that the stochastic gradient steps consider contributions from both enantiomers. In both cases, we average the loss contributions for all conformers in the batch. In the case of ChIRo, we add the mean auxiliary torsion loss across all conformers in the batch to the batch cross-entropy loss.

We use the Adam optimizer with a flat learning rate throughout training, but employ gradient clipping with a maximum gradient L2-norm of 10. We train each model for a maximum of 100 epochs, and use the batch-averaged *classification accuracy* on the validation set to select the model with the best estimated generalization performance across the 100 epochs. We do not employ dropout or other forms of explicit regularization.

For testing, we evaluate the classification accuracy on *all* conformers in the test set, and do not use any form of conformer-based averaging or voting.

### A.5.7 RANKING ENANTIOMERS BY DOCKING SCORES

For ranking enantiomers by their docking scores, we train each model with a mean squared error (MSE) loss, with the target values being the ground-truth docking scores for each enantiomer. Note that although we train the models with an MSE loss, we are more concerned with the ability of each model to correctly rank enantiomers by their docking scores than with the absolute performance of each model as a surrogate model for docking score prediction. Thus, we *evaluate* this task using the ranking accuracy, defined as whether or not the model correctly predicts a lower docking score for the enantiomer with the lower ground-truth score.

In each training epoch, we randomly partition the training data into $N/b$ minibatches, where $N$ is the number of *enantiomers* in the training set and $b$ is the batch size. For each enantiomer in the batch, we sample either 1) a random conformer *or* 2) a pre-selected conformer for that enantiomer, *and* either 1) a random conformer *or* 2) a pre-selected conformer conformer for its opposite enantiomer with the same 2D graph. This ensures that minibatches contain both enantiomers of the enantiomeric pair such that the stochastic gradient steps consider contributions from both enantiomers. In both cases, we average the loss contributions for all conformers in the batch. In the case of ChIRo, we add the mean auxiliary torsion loss across all conformers in the batch to the batch MSE loss.

We use the Adam optimizer with a flat learning rate throughout training, but employ gradient clipping with a maximum gradient L2-norm of 10. We train each model for a maximum of 150 epochs,

and use the batch-averaged *ranking accuracy* on the validation set to select the model with the best estimated generalization performance across the 150 epochs. We do not employ dropout or other forms of regularization.

For testing, we first predict the docking score for *all* conformers in the test set. We then average the predicted docking scores for each conformer of each enantiomer, yielding a mean predicted score for each enantiomer. Finally, we compute the ranking accuracy using these mean predicted scores.

Table 10: Hyperparameter search space for ChIRo

| Hyperparameter | Search Space |
|---|---|
| EConv MLP Hidden Layer Size | [32, 64, 128, 256] |
| EConv MLP # Hidden Layers | [1,2] |
| $\mathbf{h}_0, \mathbf{h}_T, \mathbf{h}_t^{\text{CMP}}$ Dimension | [8, 16, 32, 64] |
| $\mathbf{h}_t, t = 1, ..., T - 1$ Dimension | [16, 32, 64] |
| # GAT Layers | [2,3,4] |
| # GAT Heads | [1,2,4,8] |
| $f_d, f_\phi, f_\alpha, f_c$ Hidden Layer Size | [32, 64, 128, 256] |
| $f_d, f_\phi, f_\alpha, f_c$ # Hidden Layers | [1,2,3,4] |
| $f_\varphi$ Hidden Layer Size | [32, 64, 128, 256] |
| $f_\varphi$ # Hidden Layers | [1,2,3,4] |
| $f_{\text{out}}$ Hidden Layer Size | [32, 64, 128, 256] |
| $f_{\text{out}}$ # Hidden Layers | [1,2,3,4] |
| $z_d, z_\phi, z_\alpha$ Dimension | [8, 16, 32, 64] |
| CMP EConv Hidden Layer Size | [32, 64, 128, 256] |
| CMP EConv # Hidden Layers | [1,2,3,4] |
| # CMP GAT Layers | [1, 2, 3, 4] |
| # CMP GAT Heads | [1, 2, 4, 8] |
| $\gamma_{\text{aux}}$ | log uniform (1e-4, 1e-2) |
| Learning Rate | log uniform (5e-5, 5e-3) |
| Batch Size | [16, 32, 64, 128, 256] |

## A.6 HYPERPARAMETER OPTIMIZATIONS

### A.6.1 CHIRO

Hyperparameters were tuned for ChIRo on the R / S, $l$ / $d$, and ranking enantiomers by docking score tasks, using the Raytune (Liaw et al., 2018) Python package with the HyperOpt plug-in. For the R / S classification and ranking enantiomers by docking score tasks, we tuned hyperparameters by training with the training set and evaluating model accuracy on the validation set. For the $l$ / $d$ classification task, we used the training and validation splits of the first cross-validation fold to tune hyperparameters, evaluating model accuracy on the validation split. The optimal hyperparameters were then held constant for the remaining four folds.

For each task below, we used the HyperOptSearch search algorithm in Raytune, which employs Tree-structured Parzen Estimators, to search for optimal hyperparameters over the search space specified in Table 10.

**R / S Classification.** We trained a total of 100 models with different hyperparameter combinations for a maximum of 50 epochs, using the Async Hyperband Scheduler (grace period = 5, reduction factor = 3, brackets = 1) for aggressive early stopping.

**$l$ / $d$ Classification** We performed two stages of optimization. We first trained 100 models with different hyperparameter combinations for a maximum of 50 epochs, using the Async Hyperband Scheduler (grace period = 5, reduction factor = 3, brackets = 1) for aggressive early stopping. We then re-ran the optimization for 50 parameter combinations over a maximum of 100 epochs, using the best five models from the first optimization as starting (seed) configurations. In this second stage, we again used HyperOptSearch but changed the Async Hyperband Scheduler to use parameters (grace period = 10, reduction factor = 4, brackets = 1).

**Ranking enantiomers by docking score.** We trained a total of 100 models with different hyperparameter combinations for a maximum of 50 epochs, using the Async Hyperband Scheduler (grace period = 5, reduction factor = 3, brackets = 1) for aggressive early stopping.

Table 11: Hyperparameter search space for SphereNet

| Hyperparameter | Search Space |
|---|---|
| hidden_channels | [64, 128, 256] |
| out_channels | [16, 32, 64, 128, 256] |
| cutoff | [5.0, 10.0] |
| num_layers | [3,4,5] |
| int_emb_size | [32, 64, 128] |
| basis_emb_size_dist | 8 |
| basis_emb_size_angle | 8 |
| basis_emb_size_torsion | 8 |
| out_emb_channels | [64, 128, 256] |
| num_spherical | 7 |
| num_radial | 6 |
| envelope_exponent | 5 |
| num_before_skip | 1 |
| num_after_skip | 2 |
| num_output_layers | 3 |
| Readout MLP Hidden Size | [32, 64, 128, 256] |
| Readout MLP # of Hidden Layers | [1,2,3,4] |
| Learning Rate | log uniform (5e-5, 1e-2) |
| Batch Size | [16, 32, 64, 128, 256] |

Table 12: Hyperparameter search space for DMPNN

| Hyperparameter | Search Space |
|---|---|
| hidden_size | [300, 600, 900, 1200] |
| depth | [2, 3, 4, 5, 6] |
| dropout | [0, 0.2, 0.4, 0.6, 0.8, 1] |
| Max Learning Rate | log uniform (1e-5, 1e-3) |
| Batch Size | [25, 50, 100] |

### A.6.2    SPHERENET

Hyperparameters were also tuned for SphereNet on the R / S, $l$ / $d$, and ranking enantiomers by docking score tasks, using the same optimization methodologies as when tuning ChIRo. The hyperparameter search space is specified in Table 11. In each of these tasks, we trained a total of 50 models with different hyperparameter combinations for a maximum of 50 epochs, using the Hyper-OptSearch algorithm and the Async Hyperband Scheduler (grace period = 5, reduction factor = 3, brackets = 1) for aggressive early stopping.

### A.6.3    DMPNN

We tune hyperparameters for the DMPNN on the $l$ / $d$ classification and ranking enantiomers by docking score tasks using the original code provided by Pattanaik et al., which employs the Optuna hyperoptimization framework (Akiba et al., 2019). We train a total of 100 models using the HyperbandPruner algorithm to prune unpromising trials and the CmaEsSampler to sample new trials. Note that we only optimize hyperparameters for the sum aggregation baseline model, and we extend these hyperparameters to the TetraDMPNN models, following the work of Pattanaik et al.. Because the TetraDMPNN models use a permutation-based aggregation function, the runtime of these models is much slower, which renders a full hyperparameter optimization infeasible.

## A.7   DATASETS

### A.7.1   CONTRASTIVE LEARNING AND R/S CLASSIFICATION DATASETS

To create the datasets for the contrastive learning and R/S classification tasks, we first randomly selected 50 sdf files from the "10 conformers per compound" directory from the PubChem3D FTP site ftp://ftp.ncbi.nlm.nih.gov/pubchem/Compound_3D/, out of 6199 available. After extracting the conformers in each sdf file, we filtered the data to only include conformers of 2D graphs for which *at least* 2 stereoisomers appear in the data, and where each of these stereoisomers has *at least* 2 conformers in the data. We further filtered the data to only include 2D graphs whose conformers contain at least 1 torsion, as recognized by RDKit. This dataset was separated into 80/20 partitions, with conformers corresponding to the same 2D graph being assigned to the same data partition. The larger (80%) subset was used as-is for contrastive learning. The smaller (20%) subset was further filtered to only include pairs of enantiomers with *only* 1 chiral center, excluding 2D graphs that contained other (cis/trans) stereoisomers of these enantiomer pairs in the dataset. Note that this extra step ensures that there are only two stereoisomers (which are enantiomers) per 2D graph. The resultant subset was used as the R/S classification dataset.

Figures 5 and 6 show the distributions of the number of conformers per stereoisomer and the number of stereoisomers per 2D graph in the full contrastive learning dataset. Figure 7 shows the distribution of the number of conformers per enantiomer in the full R/S classification dataset. Table 13 indicates the distribution of R/S labels in the R/S dataset.

The full contrastive learning dataset was split into 70/15/15 sets for training, validation, and testing, respectively, with conformers corresponding to the same 2D graphs being assigned to the same data partition. The training set contains 2,088,008 conformers of 418,922 stereoisomers of 180,426 distinct 2D graphs. The validation set contains 450,726 conformers of 89,786 stereoisomers of 38,658 distinct 2D graphs. The test set contains 448,017 conformers of 89,914 stereoisomers of 38,659 distinct 2D graphs.

The full R/S dataset was similarly separated into 70/15/15 sets, with pairs of enantiomers being assigned to the same data partition. The training set contains 326,865 conformers of 55,084 stereoisomers (27,542 pairs of enantiomers). The validation set contains 70,099 conformers of 11,748 stereoisomers (5874 pairs of enantiomers). The test set contains 69,719 conformers of 11,680 stereoisomers (5840 pairs of enantiomers).

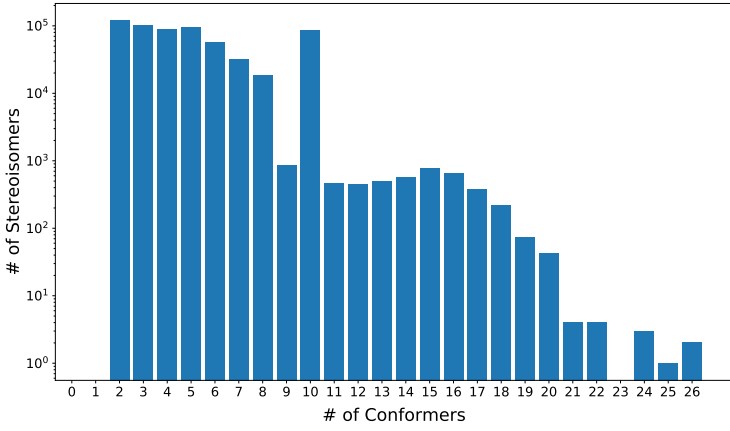

Figure 5: Histogram of the number of conformers per stereoisomer in the contrastive learning dataset.

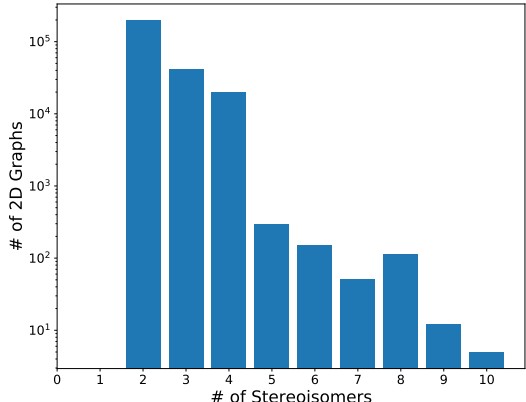

Figure 6: Histogram of the number of stereoisomers per 2D graph in the contrastive learning dataset.

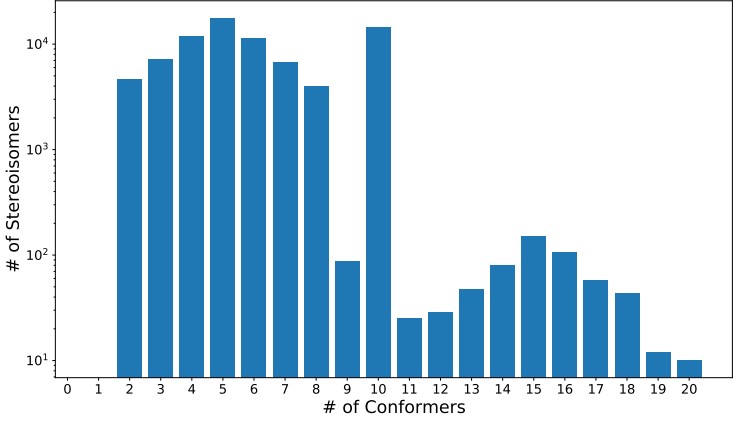

Figure 7: Histogram of the number of conformers per enantiomer in the R/S classification dataset.

Table 13: Balance of R/S labels in the R/S classification dataset.

| R / S Label | # of Conformers | # of Enantiomers |
|:---:|:---:|:---:|
| R | 236222 | 39256 |
| S | 230461 | 39256 |

### A.7.2 $l/d$ CLASSIFICATION DATASET

Figure 8 enumerates the data filtering steps used to extract and filter the $l/d$ classification dataset from the experimental Reaxys database. In Reaxys, we queried non-fragmented molecules with molecular weights $\leq 564$ which had optical rotatory power reported at 18-30 $^{o}$C and a wavelength of 589 nm. We further filtered these molecules to include those with SMILES strings that were validated by RDKit, and those which contained only 1 chiral center. Of these molecules, we removed duplicate entries if they were reported with different signs of optical rotation (which indicates an experimental measurement error). If duplicates had consistent signs of optical rotation, we randomly selected one entry. We then removed entries that did not have full stereochemistry specified in their SMILES strings, since we would later need to generate conformers for each molecule. We also removed molecules whose non-enantiomeric (e.g., cis/trans) stereoisomers also appeared in the dataset. We then checked if pairs of enantiomers were both reported in the dataset, and excluded such pairs if they were reported with the same sign of optical rotation (which indicates an experimental measurement error). For molecules whose opposite enantiomers were not in the dataset, we artificially generated the opposite enantiomer and assigned it the opposite sign of optical rotation. This ensures a balanced dataset. We then selected pairs of enantiomers which were reported with $\geq$ 95% enantiomeric excess in order to reduce the risk of experimental noise causing labeling errors. Lastly, we used RDKit to generate 5 conformers for each enantiomer in the filtered dataset. If RDKit failed to generate a conformer for any pair of enantiomers, both enantiomers in the pair were removed from the dataset.

Table 14 reports the distribution of $l/d$ labels in the final dataset, which contains 150,380 conformers of 30,076 enantiomers (15,038 pairs of enantiomers). We split this dataset into 5 folds for cross-validation, randomly assigning each pair of enantiomers (with their conformers) to a test set in one of the five folds. For each fold, we randomly split the remaining (i.e., non-testing) pairs of enantiomers into 87.5/12.5 training/validation sets. Note that this ensures each fold has 70/10/20 training/validation/test splits, with pairs of enantiomers being assigned to the same data partition within each fold, and that each pair of enantiomers only appears in one test set across the five folds.

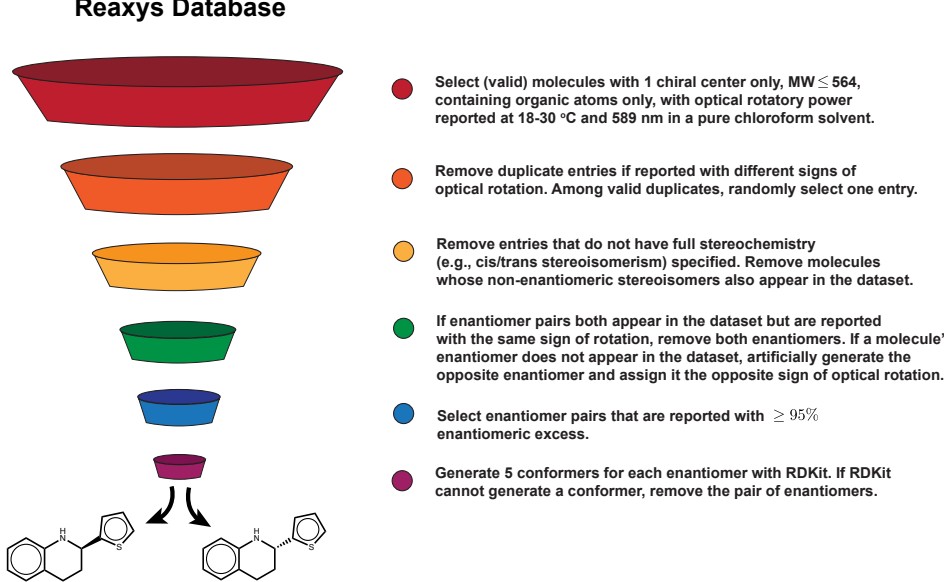

Figure 8: Data extraction, filtering, and generation steps for the $l/d$ classification task.

Table 14: Balance of $l$ / $d$ labels in dataset.

| $l$ / $d$ Label | # of Conformers | # of Enantiomers |
|:---:|:---:|:---:|
| $l$ | 75190 | 15038 |
| $d$ | 75190 | 15038 |

Table 15: Frequency of R/S labels amongst enantiomers in the $l$ / $d$ dataset that rotate light in the positive ($d$) or negative ($l$) direction. The balance in R/S labels indicates the lack of empirical correlation between these two classification schemes.

| | $l$ | $d$ |
|:---:|:---:|:---:|
| **R** | 7316 | 7722 |
| **S** | 7722 | 7316 |

### A.7.3 RANKING ENANTIOMERS BY DOCKING SCORES

To create the dataset for the ranking enantiomers by their docking scores task, we randomly selected 750 sdf files from the "10 conformers per compound" directory from the PubChem3D FTP site `ftp://ftp.ncbi.nlm.nih.gov/pubchem/Compound_3D/`, out of the 6199 available, but excluding the 50 sdf files used previously for the contrastive learning and R/S classification datasets. After extracting the conformers in each sdf file, we filtered the data to only include conformers of 2D graphs for which *only* 2 stereoisomers (corresponding to pairs of enantiomers with 1 chiral center) appeared in the dataset. As before, we only included enantiomers whose conformers contain at least 1 torsion, as recognized by RDKit. We then filtered enantiomers which have a molecular weight $\leq 225$ and $\leq 5$ rotatable bonds (as computed by RDKit). This step was designed to intentionally select small enantiomers which had few rotational degrees of freedom such that the docking simulations would be less stochastic. We then docked each pair of enantiomers three times against the protein (PDB ID: 4JBV) in the docking box centered at $[10, 16, 61]$ with (x,y,z) radii of $[10, 14, 12]$. In each docking simulation, we increased the exhaustiveness parameter to 24 to help reduce noise in the resultant docking scores. As a final control for stochasticity in the docking simulations, we also removed pairs of enantiomers for which one or both of the enantiomers had a range of (top) docking scores $> 0.1$ kcal/mol across the three simulation trials. Finally, in order to select pairs of enantiomers which exhibit meaningful differences in docking scores, we filtered the dataset to only include pairs of enantiomers which had a difference in their best docking scores (across the three trials) of at least $0.3$ kcal/mol. These enantiomers and their PubChem3D conformers were used as the final dataset. Figure 9 visualizes the full filtering procedure.

Figure 10 plots the distribution of the number of conformers per enantiomer in the docking dataset. Figure 11 plots the distribution of the difference in docking scores between pairs of enantiomers in the docking dataset. We split the full dataset into 70/15/15 training/validation/test sets, assigning pairs of enantiomers (with their conformers) to the same data partition. The training set contains 234,622 conformers of 48,384 enantiomers (24,192 pairs of enantiomers). The validation set contains 49,878 conformers of 10,368 enantiomers (5184 pairs of enantiomers). The test set contains 50,571 conformers of 10,368 enantiomers (5184 pairs of enantiomers).

Figure 9: Data generation and filtering steps for the ranking enantiomers by docking scores task.

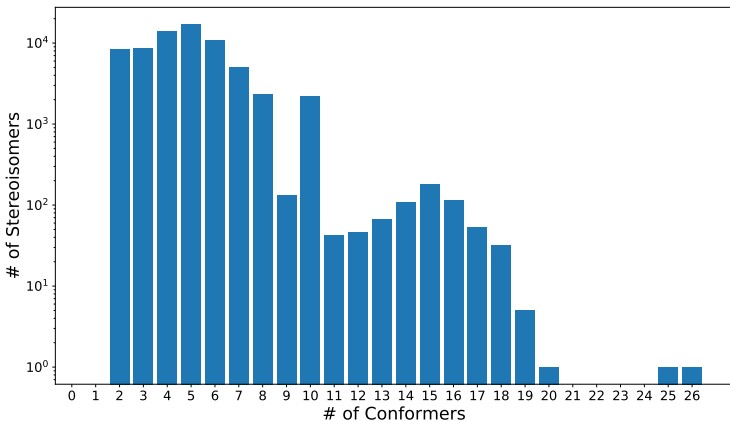

Figure 10: Histogram of the number of conformers per enantiomer in the docking dataset.

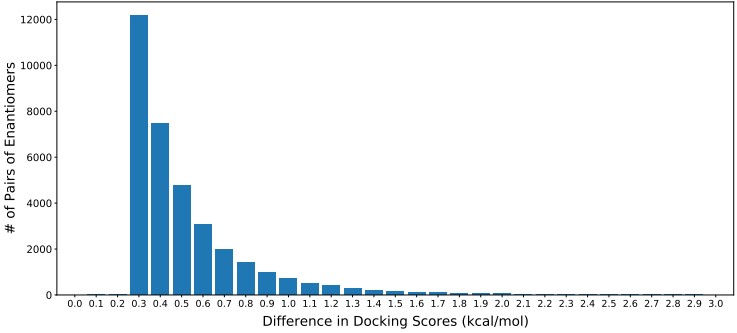

Figure 11: Distribution of the differences in (best) docking scores between pairs of enantiomers in the docking dataset.

## A.8 Additional Ablations

It is not strictly necessary that ChIRo *learn* the weight coefficients $c_{ij}^{(xy)}$ in order to distinguish enantiomers. ChIRo can still learn chiral representations and preserve invariance to internal bond rotations if each $c_{ij}^{(xy)}$ is generated by a network with random, untrained parameters. However, this is not the case for the phase shifts $\varphi_{ixyj}$, which must be learned for good performance. To demonstrate this, we perform additional ablations where the feed-forward network $f_c$ is not trained, but rather keep its initial random parameters. We perform the same analysis for $f_\varphi$ to also evaluate whether ChIRo needs to *learn* the phase shifts $\varphi_{ixyj}$, or if these can also be generated by a random network. Table 16 reports the results of not training these MLPs on the otherwise unchanged ChIRo on the $l / d$ classification task and ranking enantiomers by docking scores task. Overall, learning the parameters in $f_c$ leads to only small (if any) performance gains versus keeping the randomly initialized parameters. On the other hand, learning the parameters in $f_\varphi$ leads to considerable performance gains versus not learning $f_\varphi$. This emphasizes that in our case, relying on random noise to break symmetry between embeddings of enantiomers is insufficient to learn *expressive* chiral representations. Rather, ChIRo best models the effects of chirality by *learning* to break the symmetry through learned phase shifts in a task-specific manner.

Table 16: Effects of not learning the parameters in $f_c$ and $f_\varphi$ on ChIRo's performance on the $l / d$ classification and the ranking enantiomers by docking score tasks. For the original, unablated ChIRo (first row), mean accuracy and standard deviations on the test sets are reported across 5 folds ($l / d$) or three repeated trials (enantiomer ranking). To better account for the impact of random network initializations, for the ablated models we report the mean accuracy and standard deviations on the test sets across 3 repeated trials (enantiomer ranking) or the mean, fold-averaged accuracy (e.g., mean of means) and standard error of this mean across three repeated 5-fold CV trials ($l / d$ classification).

| Model Components | | Accuracy (%) ↑ | |
| --- | --- | --- | --- |
| Learned $f_c$ | Learned $f_\varphi$ | $l / d$ | Enantiomer Ranking |
| ✓ | ✓ | $79.3 \pm 0.4$ | $72.0 \pm 0.5$ |
| X | ✓ | $79.4 \pm 0.3$ | $71.7 \pm 0.2$ |
| ✓ | X | $53.7 \pm 1.2$ | $68.4 \pm 0.7$ |
| X | X | $50.8 \pm 0.8$ | $66.8 \pm 0.3$ |

## A.9 Additional Evaluation of Ranking Enantiomers by Docking Scores

Because some enantiomers have larger differences in their ground truth docking scores than other enantiomers, a single ranking accuracy metric may not fully describe the ability of the models to learn the enantioselectivity of the protein pocket. To evaluate model performance more thoroughly, we compute ranking accuracies on various subsets of the test set. Figure 12 plots the ranking accuracies of ChIRo, SphereNet, Tetra-DMPNN (concatenate) with chiral tags, and DMPNN with chiral tags when evaluated on subsets of the test data where the difference in ground truth docking scores are (upper left) greater or equal to a margin, (upper right) less than or equal to a margin, or (bottom) exactly equal to a margin. The plots suggest that while ChIRo is superior in correctly ranking enantiomers which have relatively small differences in their true docking scores, the differences between each model become less distinct when ranking enantiomers with larger differences in their ground-truth docking scores. This may be due to the fact that the docking dataset is imbalanced, skewed heavily toward enantiomers that have smaller differences in their true docking scores (Figure 11).

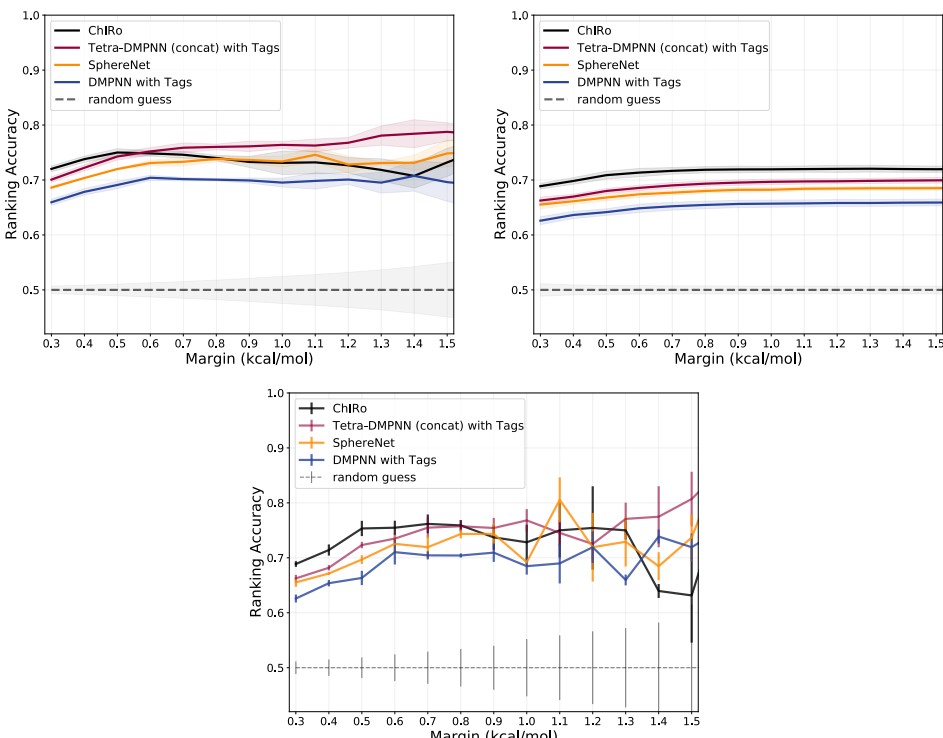

Figure 12: Ranking accuracy of the predicted conformer-averaged docking scores when evaluated on subsections of the test set in which the ground truth difference in best docking scores between enantiomers is (upper left) greater than or equal to the specified margin, (upper right) less than or equal to the specified margin, or (bottom) equal to the specified margin. Error bars for the models represent standard deviations in ranking accuracy across three random training/inference trials. Error bars for the random baseline correspond to the standard deviation of a binomial distribution $B(p = 0.5, N)$, where $N$ is the number of enantiomer pairs in the subset.

### A.10 ANALYZING SE(3)-INVARIANT 3D GNNS WITHOUT INTERROTO-INVARIANCE

The qualitative results shown in Figure 3 suggest that an SE(3)-invariant 3D GNN without InterRoto-invariance, such as SphereNet, will get confused between differences in molecular chirality (e.g., inverted chiral centers) with differences in conformational structure (e.g., bond rotations) when learning chiral-dependent functions. To support this hypothesis with quantitative evidence, Figure 13 shows the distribution of the fraction of conformers per enantiomer in the test sets of the $l/d$ classification task which were predicted to have the same sign of optical rotation by SphereNet and ChIRo, irrespective of the actual accuracy of the predicted class labels. SphereNet predicts the same label for each of the 5 RDKit-generated conformers per enantiomer for only ~32% of the enantiomers in the test sets, compared to ~93% for ChIRo. When only trained on 1 conformer per enantiomer, SphereNet's labeling consistency drops to 14%, whereas ChIRo's labeling consistency remains roughly the same at ~92%. ChIRo's dramatically increased consistency in labeling conformers sharing the same chiral molecular identity compared to SphereNet is a direct result of ChIRo's InterRoto-invariance. Note that ChIRo's consistency is not perfectly 100% because the RDKit-generated conformers have conformational differences not associated with simple bond rotations. For instance, two conformers can slightly differ in their bond distances and bond angles, perturbations to which ChIRo is *not* invariant.

Moreover, manually rotating bonds near the chiral center (similar to the bottom row in Figure 3) causes SphereNet to get confused when predicting signs of optical rotation for these rotated conformers, even for enantiomers (in the test-set) whose conformers were all originally classified correctly by SphereNet (Figure 14). Since ChIRo is invariant to such bond rotations (Figure 3), it is not susceptible to this particular type of confusion, and therefore has increased ability to generalize to unseen chiral conformers.

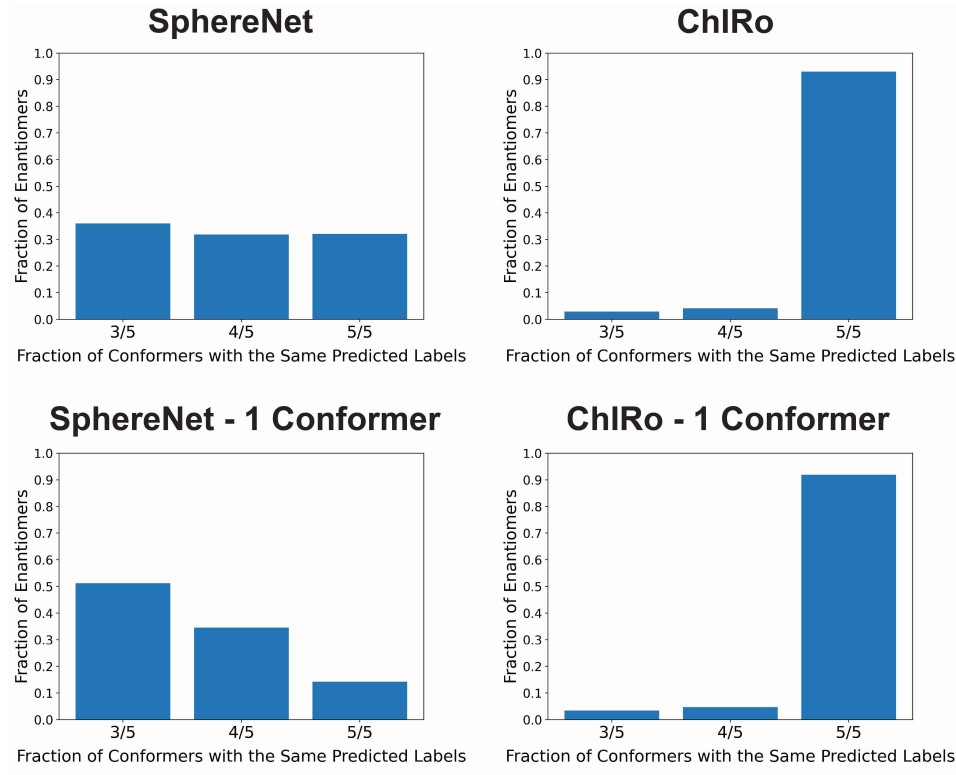

Figure 13: Distribution of the fraction of conformers per enantiomer predicted to have the same class label for the $l/d$ classification task across the test sets in each of the five folds, for both SphereNet and ChIRo when trained on all conformers in the training set (top row) or on a single conformer per enantiomer (bottom row). Note that for the $l/d$ classification task, there are exactly five conformers per enantiomer in the test splits of each fold.

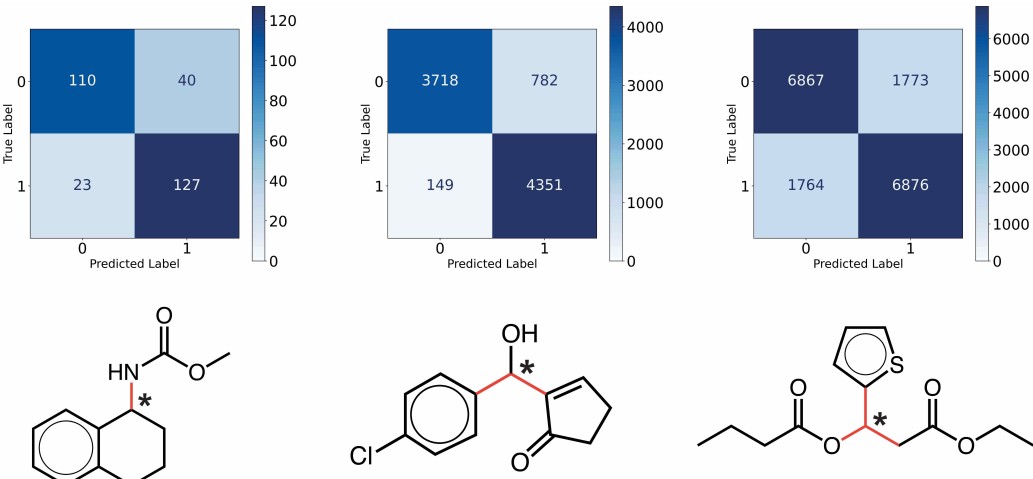

Figure 14: Confusion matrices indicating SphereNet's (mis)classification of conformers of three pairs of enantiomers in the $l/d$ classification task's test set (of the first fold) whose highlighted bonds are each rotated in increments of $12°$ (left and center) or $30°$ (right). These three pairs of enantiomers were specifically chosen for this confusion analysis because SphereNet correctly classifies *all* of their original (non-rotated) conformers.

