# OpenReview forum: "Learning 3D Representations of Molecular Chirality with Invariance to Bond Rotations"
_ICLR.cc/2022/Conference — ICLR 2022 Poster_

### Official Review · Reviewer_KqtQ · 2021-10-29

**Correctness:** 4
**Technical Novelty And Significance:** 4
**Empirical Novelty And Significance:** 3
**Recommendation:** 8
**Confidence:** 4

**Main Review:**

[Novelty]

From my point of view, if a molecular representation model wants to achieve good expressivity, it should:

1. perceive more specific molecular properties, such as chirality utilized to separate enantiomers;
2. be invariant to abundant information for better generalization.

Traditional models, such as 2D-GNNs, E(3)-Invariant ones and SE(3)-Invariant ones, can't meet the requirements above , so their expressivity is limited. Contrastively, ChIRo well tackles these two issues by defining and implementing (SE(3)+InterRoto)-Invariance and is proved to be more expressive.

[Theory]

ChIRo uses a series of trigonometric functions to encode the torsion angles. To demonstrate that such encoding is invariant to bond torsion but sensitive to chirality, this paper gives a proof in Appendix. This proof is clear but Figure 2 makes me confused before reading the proof. (This will be elaborated in Cons.[Illustration].) Despite this flaw, the theoretical foundation of this paper is very solid.

[Reproduction]

In Section 3, it gives a very clear description on the structure of ChIRo. And implementation details are presented in Appendix.

[Representation]

This paper is very clear and easy to understand.



Cons:

[Model Design]

As far as we know, most molecular representation models considering 3D structure use the spatial information when passing messages among the atoms and bonds, such as 3DGCN, DimeNet and PhysChem(https://openreview.net/forum?id=Uxi7X1EqywV). However, ChIRo consists of a 2D-GNN and an individual NN to embedding D/Phi/Psi found in molecular conformation. We think the way ConfAE (the method ChIRo follow) processes the spatial information will lose graph topology, because D/Phi/Psi are detached from message passing and treated as "bags of words". This reason of adopting such structure is "to be invariant to the order of presenting the atoms", but actually the output of MPNN is invariant to input atom order if only order-independent aggregation functions are used (such as sum/mean pooling). Therefore, we think the structure design of ChIRo can be further improved.

[Illustration]

In Figure 2, the contents in 3rd and 4th columns are confusing before the readers fully understand the method and its demonstration (e.g. what's the relationship between the solid sinusoids and dotted ones, what does "r" mean and so on). We hope the illustrations can be better arranged.

In Figure 3, the illustrations of "Original OMEGA Conformers" and "Original+Reflected Conformers" are almost the same, we can't figure out what the difference is between them.

[Experiment]

Although ChIRo is evaluated in 4 tasks, its application value remains suspicious.

Firstly, in the first 3 tasks (Contrastive Learning to Distinguish Stereoisomers, Classifying Tetrahedral Chiral Centers as R/S and Predicting Enantiomers’ Signs of Optical Rotation), the targets are strongly correlated to the chirality of enantiomers. So, these tasks seem more like sensitivity tests of "if ChIRo can actually perceive chirality information" rather than "if ChIRo can handle existing tasks".

Secondly, ChIRo is labeled as a "molecular representation model", so it's necessary to demonstrate its effectiveness on various popular tasks on molecules, such as QM7/8/9 with given conformations (see MoleculeNet(https://pubs.rsc.org/en/content/articlelanding/2018/sc/c7sc02664a)), or other molecular property prediction tasks (FreeSolv, TOX21, ...) assisted with RDKit-generated conformations (see HamNet(https://openreview.net/forum?id=q-cnWaaoUTH)). Besides the 3 tasks mentioned in last paragraph, this paper only evaluate ChIRo on one dataset of molecule-protein binding. Therefore, we think ChIRo has not yet been proved to be a SOTA molecular representation model.

Thirdly, in Ablation Study, we see "adding chiral tags to our unablated model does not improve performance". We think that's caused by overfitting to abundant tags (as chirality information has been well embedded thru torsion encoder), and we hope a validation of it can be given in this paper rather than just a statement of this abnormal phenomenon.

**Summary Of The Paper:**


This paper proposes a 3D-GNN molecular representation model  (ChIRo) which:

1. can perceive chirality information to improve expressivity;
2. is invariant to bond rotations and mitigates the need for data augmentation.

**Summary Of The Review:**

This paper proposes ChIRo to improve the expressivity in molecular representation by well tackling chirality. It has a solid theoretical foundation and a clear and relatively well-designed model structure. Although the experiment is not yet sufficient to demonstrate its application value, we believe that ChIRo's effectiveness can be claimed after being evaluated on diverse tasks on molecular representation, such as molecular property prediction datasets in MoleculeNet.

In summary, it's a good piece of work but needs some improvement.

---

> ### Author Response · Authors · 2021-11-16
> **Authors' Response to Reviewer KqtQ (Part 1)**
>
> We thank the reviewer for their comments and suggestions. We have provided detailed responses to each of their specific comments below. We ask the reviewer to consider increasing their recommendation if we fully address their concerns.
>
> _Cons:_
>
> _[Model Design]_
>
> * _As far as we know, most molecular representation models considering 3D structure use the spatial information when passing messages among the atoms and bonds, such as 3DGCN, DimeNet and PhysChem(https://openreview.net/forum?id=Uxi7X1EqywV). However, ChIRo consists of a 2D-GNN and an individual NN to embedding D/Phi/Psi found in molecular conformation. We think the way ConfAE (the method ChIRo follow) processes the spatial information will lose graph topology, because D/Phi/Psi are detached from message passing and treated as "bags of words". This reason of adopting such structure is "to be invariant to the order of presenting the atoms", but actually the output of MPNN is invariant to input atom order if only order-independent aggregation functions are used (such as sum/mean pooling). Therefore, we think the structure design of ChIRo can be further improved._
>
> This is an interesting consideration, and one that the authors also considered in Appendix A.3. To summarize here, we considered treating each internal bond embedding ${{z}}\_{\alpha\_{xy}}$ (constructed from Equation 12) as a vector of edge attributes that could be propagated across the graph through an additional round of message passing. However, our so-called “Chiral Message Passing” scheme did not improve the performance of the base ChIRo model. We believe it is an open question on how to best propagate and pool chiral information across the graph to obtain better global representations of molecular chirality.
>
> One justification for using individual neural networks to encode the dihedrals is to make ChIRo modular and insertable into existing 2D GNN frameworks in order to easily augment their ability to learn tetrahedral chirality, while still maintaining InterRoto-invariance. This is why we also used standard off-the-shelf 2D GNNs (e.g., EConv/GAT) to encode the node states, rather than over-optimizing factors we do not consider to be the core novel contributions of ChIRo.
>
> _[Illustration]_
>
> * _In Figure 2, the contents in 3rd and 4th columns are confusing before the readers fully understand the method and its demonstration (e.g. what's the relationship between the solid sinusoids and dotted ones, what does "r" mean and so on). We hope the illustrations can be better arranged._
>
> The visualization in Figure 2 is meant to complement the mathematical methods outlined in Equations 10-11, e.g. to provide a visual geometric interpretation of how our method achieves InterRoto-invariance and chiral expressivity. To help clarify aspects of Figure 2, we have added clarifying statements in the accompanying caption.
>
> * _In Figure 3, the illustrations of "Original OMEGA Conformers" and "Original+Reflected Conformers" are almost the same, we can't figure out what the difference is between them._
>
> “Original+Reflected Conformers” includes _additional_ points that correspond to the reflected conformers in “Original OMEGA Conformers”. Note that these reflected conformers will have inverted colors (blue to red, and vice versa) because reflecting conformers inverts their chirality. ChIRo’s latent space visualization is unchanged, since ChIRo perfectly clusters  enantiomers due to its InterRoto-invariance. SphereNet’s latent space is also largely unchanged, but you can count that there are additional points in each cluster, corresponding to the additional reflected conformers (which are still clustered correctly). SchNet and DimeNet++ are E(3)-invariant, and thus are invariant to reflections. As a result, the points have become _purple_ (rather than red/blue) because the reflected conformers perfectly overlap the original conformers.

---

> > ### Author Response · Authors · 2021-11-16
> > **Authors' Response to Reviewer KqtQ (Part 2)**
> >
> > _[Experiment]_
> >
> > _Although ChIRo is evaluated in 4 tasks, its application value remains suspicious._
> >
> > * _Firstly, in the first 3 tasks (Contrastive Learning to Distinguish Stereoisomers, Classifying Tetrahedral Chiral Centers as R/S and Predicting Enantiomers’ Signs of Optical Rotation), the targets are strongly correlated to the chirality of enantiomers. So, these tasks seem more like sensitivity tests of "if ChIRo can actually perceive chirality information" rather than "if ChIRo can handle existing tasks"._
> >
> > The tasks considered in our submission are designed to 1) highlight the limitations of existing 3D and 2D GNNs when learning the effects of tetrahedral chirality, and 2) empirically highlight ChIRo’s superior expressivity and generalizability in this regard. We have not explicitly designed ChIRo, in its current construction, as a new SOTA general-purpose molecular property predictor. Instead, we have designed ChIRo to be modular and able to augment existing (and future) SOTA 2D GNNs to improve their chiral expressivity. In our submission, we used EConv/GAT layers as the 2D GNN, but in principle any current SOTA 2D model could be plugged in.
> >
> > As a corollary to the above, because we include the pooled 2D node states in our readout-phase, ChIRo will not restrict any 2D model’s ability to handle property prediction tasks. Thus, applying ChIRo to existing tasks (that are not chiral-dependent, which includes most common benchmarks) is akin to evaluating standard GNNs on these tasks, which is neither interesting nor novel. We have explicitly designed our tasks to introduce new benchmarks which are sensitive to chirality, thus shedding new light on existing models’ abilities to learn and express tetrahedral chirality.
> >
> > * _Secondly, ChIRo is labeled as a "molecular representation model", so it's necessary to demonstrate its effectiveness on various popular tasks on molecules, such as QM7/8/9 with given conformations (see MoleculeNet(https://pubs.rsc.org/en/content/articlelanding/2018/sc/c7sc02664a)), or other molecular property prediction tasks (FreeSolv, TOX21, ...) assisted with RDKit-generated conformations (see HamNet(https://openreview.net/forum?id=q-cnWaaoUTH)). Besides the 3 tasks mentioned in last paragraph, this paper only evaluate ChIRo on one dataset of molecule-protein binding. Therefore, we think ChIRo has not yet been proved to be a SOTA molecular representation model._
> >
> > Please see the immediately preceding statements, which directly address these concerns. We also direct the reviewer  to our responses to Reviewer 3 on the applicability of ChIRo to the QM9 dataset.
> >
> > * _Thirdly, in Ablation Study, we see "adding chiral tags to our unablated model does not improve performance". We think that's caused by overfitting to abundant tags (as chirality information has been well embedded thru torsion encoder), and we hope a validation of it can be given in this paper rather than just a statement of this abnormal phenomenon._
> >
> > We argue that our submission has already made clear that chiral atom tags are less expressive toward tetrahedral chirality than ChIRo’s processing of the 3D torsion angles. We specifically point to Table 2, where the directed message passing neural network (DMPNN/ChemProp) with chiral atom tags, a well-established 2D GNN achieving around SOTA in chemical property prediction tasks, performs 5 - 6% worse than ChIRo when predicting enantiomers’ signs of optical rotation and ranking enantiomers by their docking scores in a chiral protein pocket. We also point to the second to last row in Table 3, which evaluates the ability of our 2D GNN (EConv/GAT) with chiral tags to learn the effects of chirality in comparison to the unablated ChIRo. Here, EConv/GAT with chiral atom tags performs 8 - 9% worse than ChIRo on the same tasks. Thus, we are not surprised that adding chiral tags to the unablated ChIRo does not improve performance--it would be strange if it did, as that would imply a curious synergistic effect.
> >
> > In addition, one might consider that ChIRo was explicitly designed assuming that the node states, embedded with a 2D GNN, are invariant to tetrahedral chirality. This assumption explicitly motivated the inclusion of learned phase shifts, which produce differing degrees of wave interference between inverted tetrahedral chiral centers when summing the weighted sines and cosines of the coupled dihedrals in Equations 10-11. If we include chiral atom tags as node features, then the learned phase shifts (and the weighting coefficients) are no longer guaranteed to be equivalent between enantiomers (as their 2D graphs are no longer the same), and thus the underlying design principles of ChIRo no longer strictly apply.

---

> > ### Comment · Reviewer_KqtQ · 2021-11-21
> > **Statisfied with Author's Detailed Reponse**
> >
> > Thank you for your clarification on [Model Design], [Illustration] and [Experiment]. Now we believe that ChIRo is an excellent work and update RECOMMENDATION and CONFIDENCE.
> >
> > [Model Design]
> >
> > > However, our so-called “Chiral Message Passing” scheme did not improve the performance of the base ChIRo model.
> >
> > This is a very interesting question. Since detaching chirality information from message passing can still achieve good performance similar to using “Chiral Message Passing”, your model design seems not a problem.
> >
> > [Experiment]
> >
> > > The regression targets in QM9 are invariant to tetrahedral chirality, since quantum mechanical energies for single, non-interacting molecules are invariant to the reflection transform.
> >
> > We agree with you that ChIRo is designed to *empirically highlight ChIRo’s superior expressivity and generalizability*, so it can't handle the most prediction tasks on quantum property e.g. QM9. In this case, evaluating ChIRo on chirality-sensitive tasks only seems more reasonable.
> >
> > However, we still feel not that confident of ChIRo's application value, because ChIRo is evaluated on only one molecule-protein binding dataset besides the tasks designed by yourself. We hope that there are more tests on diverse molecule-protein binding datasets or other popular molecular representation tasks which is chirality-sensitive.

---

> > > ### Author Response · Authors · 2021-11-22
> > > **Follow-up to Reviewer KqtQ**
> > >
> > > Thank you for increasing your recommendation and for your characterization that "ChIRo is an excellent work".
> > >
> > > As for the last comment:
> > > > However, we still feel not that confident of ChIRo's application value, because ChIRo is evaluated on only one molecule-protein binding dataset besides the tasks designed by yourself. We hope that there are more tests on diverse molecule-protein binding datasets or other popular molecular representation tasks which is chirality-sensitive.
> > >
> > > We direct the reviewer to our "Clarifications addressed to Reviewer Viqv", which addresses the difficulty of using popular molecular representation datasets, such as those in MoleculeNet, to benchmark the ability of models to express chirality. To reiterate here, because of the noise in these experimental datasets, it is difficult to meaningfully learn the nuanced effects of chirality. This is especially true if the dataset does not contain both enantiomers of a pair (or multiple stereoisomers in general), which is often the case. We have specially designed our datasets and tasks to 1) mitigate this experimental noise, and 2) always include both enantiomers. This is summarized in our Appendices A.7.2 and A.7.3.

---

### Official Review · Reviewer_3jRF · 2021-11-02

**Correctness:** 4
**Technical Novelty And Significance:** 3
**Empirical Novelty And Significance:** 3
**Recommendation:** 8
**Confidence:** 5

**Main Review:**


**Strength**

1. Good efforts towards incorporating inductive bias from science perspectives into model designs, have a quite lot insights.

**Weakness**

1. Rationale of the problem setting. Base on my experiments (using the models and data from [1]), using "chiral atom tags" with the vanilla GIN model will reach 100% test accuracy. So it would be nice to elaborate why "omit chiral atom tags and bond stereochemistry tags"(page 4) is a rational setting.
2. For the last part of "Torsion Encoder", I understand that it is complete to include necessary independent torsion angles, but how would the model perform if we feed it more, redundant and self-consistent features? It would be helpful to check on this :)


**Comments**

1. Why the scales of the embeddings are quite different in Figure 3 (1e^2 to 1e^8)?
2. I feel the term "contrastive learning" more close to "instance discriminative" instead of triplet loss, it is a bit appropritate forr the term usage. If you have some literatures to back this, please let me know.

**Open Questions**

Not addressing the following comments won't have negative effects on my rating. But it would be nice if the authors can make a few clarifications, which also helps with my understandings!

1. Have you tried SE(3)-equivariant models (Fuchs et al., 2020; Thomas et al., 2018)? It is a bit sceptical that they can actually distinguish this geometry chirality (i.e, considering coupled torsions in the message updates).
2. Can it be incorporated into other models as a simple add-on/module?
3. Do improving model's capacity on chirality identification have positive impacts on the performance on benchmarks such as QM9?

**Reference**:

[1] "Message passing networks for molecules with tetrahedral chirality"
[2]

**Summary Of The Paper:**

The paper focuses on improving the capacity of GNN models, using chirality identification as the case study. As an important character to represent the geometry in molecules, chirality has a fundamental impact on the molecule properties and downstream applications. It is a major extension of [1]. The distinguishing features are mainly based on the sinusoid transformations of torsion angles, which are invariant to the bond rotations.

**Summary Of The Review:**

This paper is well written and focuses on a simple yet important problem. I tend to accept and am happy to adjust my assessment based on the rebuttal.

---

> ### Author Response · Authors · 2021-11-16
> **Authors' Response to Reviewer 3jRF (Part 1)**
>
> We thank the reviewer for their comments and suggestions. We have provided detailed responses to each of their comments below. We ask the reviewer to consider increasing their recommendation if we fully address their concerns.
>
> _Weakness_
> * _Rationale of the problem setting. Base on my experiments (using the models and data from [1]), using "chiral atom tags" with the vanilla GIN model will reach 100% test accuracy. So it would be nice to elaborate why "omit chiral atom tags and bond stereochemistry tags"(page 4) is a rational setting._
>
> [1] _"Message passing networks for molecules with tetrahedral chirality"_
>
> Perhaps the reviewer could clarify what experiments they refer to when achieving “100% test accuracy” with chiral atom tags. If this is just an R/S classification task (which is also considered in [1]), we are not at all surprised that the reviewer observes such high accuracy. Chiral atom tags include the R/S designations themselves, and thus including these tags in a model’s featurization and then testing the model’s ability to classify chiral centers as R/S is akin to testing whether the model can learn an identity operation. For this reason, we view this R/S classification task as only a simple test of whether the model can _superficially_ distinguish chiral centers. As evidenced by our experiments in Table 2, ChIRo, which in its native form does not employ chiral tags, significantly outperforms a 2D GNN (DMPNN) with chiral tags when learning non-trivial, biochemically-relevant functions of molecular chirality. We also point to the second to last row in Table 3 (our ablation study), which evaluates the ability of our 2D GNN (EConv/GAT) with chiral tags to learn the effects of chirality in comparison to the unablated ChIRo without chiral tags. Here, EConv/GAT with chiral atom tags performs 8 - 9% worse than ChIRo on the same tasks. We thus claim that such chiral tags are less powerful representations of molecular chirality compared to ChIRo’s 3D torsion encodings.
>
> We purposefully omit chiral atom tags in ChIRo to highlight the expressivity of our torsion encoder. We do test whether including chiral atom tags improves ChIRo’s performance in our ablation study (Table 3), but we find no improvement and actually slightly decreased performance. This suggests  that chiral atom tags are strictly less powerful representations of chirality compared to ChIRo’s method of encoding 3D torsion angles.
>
> * _For the last part of "Torsion Encoder", I understand that it is complete to include necessary independent torsion angles, but how would the model perform if we feed it more, redundant and self-consistent features? It would be helpful to check on this :)_
>
> ChIRo already uses every (redundant) torsion in the conformer that is formed from a non-terminal bond. The redundant torsions arise from the coupled torsions (e.g., torsions that co-vary with a rotation of a non-terminal bond), which we explicitly make use of in Equations 10-11 to learn chirality. Equation 12, which sums over all the non-terminal bonds in the conformer, therefore implicitly uses the redundant torsions. We note that explicitly encoding torsions (for instance, in Equation 12) would potentially violate the InterRoto-invariance, which emerges from our special consideration of redundant torsions in Equations 10-11.
>
> Please let us know if we have not correctly interpreted the reviewer’s suggestions.

---

> > ### Author Response · Authors · 2021-11-16
> > **Authors' Response to Reviewer 3jRF (Part 2)**
> >
> > _Comments_
> >
> > * _Why the scales of the embeddings are quite different in Figure 3 (1e^2 to 1e^8)?_
> >
> > The scales of the _unnormalized_ latent space (plotted in Figure 3) for the contrastive learning experiment are not explicitly regularized because we use a _normalized_ Euclidean distance metric in our triplet loss (Equation 14). We interpret the different scales as arising due to SchNet, DimeNet++, and SphereNet’s difficulty in clustering 3D conformers according to their chirality, compared to ChIRo. We emphasize the qualitative nature of this evaluation, whose patterns are also clearly present in the quantitative results of Tables 1-2, as well as in the new experiments in Appendix A.10.
> >
> > * _I feel the term "contrastive learning" more close to "instance discriminative" instead of triplet loss, it is a bit appropritate forr the term usage. If you have some literatures to back this, please let me know._
> >
> > We suppose this is just terminology overlap in the existing literature. While our “contrastive learning” task is related to “instance discrimination” in the sense that the task attempts to discriminate conformers of different stereoisomers, it is also a form of contrastive learning in our current formulation, as positive pairs (conformers sharing the same chiral identity) are pulled together, whereas negative pairs (conformers with different chiral identities) are pushed apart in the latent space. This closely follows the typical formulation of contrastive learning, e.g. as presented in Khosla et al. (2020) (https://arxiv.org/abs/2004.11362): “In recent years, a resurgence of work in contrastive learning has led to major advances in self-supervised representation learning. The common idea in these works is the following: pull together an anchor and a ‘positive’ sample in embedding space, and push apart the anchor from many ‘negative’ samples.”
> >
> > We achieve this clustering/separation/discrimination via a triplet loss, which is a special case of more complicated contrastive losses, yet is still typically included under the umbrella of contrastive learning losses: “Modern batch contrastive approaches subsume or significantly outperform traditional contrastive losses such as triplet, max-margin and the N-pairs loss” (Khosla et al., 2020).
> >
> > We note that our same learning objective/task (distinguishing conformers of different stereoisomers while clustering conformers with the same chiral molecular identity) could be achieved with other (more complicated) contrastive losses.

---

> > > ### Author Response · Authors · 2021-11-16
> > > **Authors' Response to Reviewer 3jRF (Part 3)**
> > >
> > > _Not addressing the following comments won't have negative effects on my rating. But it would be nice if the authors can make a few clarifications, which also helps with my understandings!_
> > >
> > > The authors are happy to address any clarifying questions.
> > >
> > > * _Have you tried SE(3)-equivariant models (Fuchs et al., 2020; Thomas et al., 2018)? It is a bit sceptical that they can actually distinguish this geometry chirality (i.e, considering coupled torsions in the message updates)._
> > >
> > > We have not compared ChIRo to SE(3)-equivariant models like those cited by the reviewer. This is due to 1) the ability of making fair comparisons between invariant and equivariant models, 2) the computational cost of training existing SE(3)-equivariant models, and 3) SE(3)-equivariant models are not InterRoto-invariant, and thus we expect them to behave similarly to SphereNet when learning tetrahedral chirality from 3D molecular structures without extensive data augmentation. Moreover, the tasks we have considered for evaluation only require an SE(3)-invariant molecular representation. It is worth noting that many SE(3)-equivariant models, when specifically applied to property prediction, use an SE(3)-invariant pooling operation in the final layer, which makes the overall model SE(3)-invariant.  We believe that a comprehensive evaluation and comparison of the abilities of 2D and 3D E(3)/SE(3) {in,equi}-variant models to express chirality is out of the scope of our submission, and perhaps better suited toward a (very interesting) dedicated review paper.
> > >
> > > * _Can it be incorporated into other models as a simple add-on/module?_
> > >
> > > Yes! ChIRo’s torsion encoding is agnostic to the 2D GNN that is used to embed the node states, and therefore can be appended to existing or future 2D GNNs to improve their ability to express tetrahedral chirality for chiral molecular property prediction tasks.  This is largely enabled by ChIRo’s InterRoto-invariance, which does not make it sensitive to the particular conformer that is provided to the model. We note that ChIRo is currently better suited toward inclusion with 2D GNNs, rather than 3D GNNs. This is because 3D GNNs embed node states with conformer/torsion-specific information, which breaks InterRoto-invariance.
> > >
> > > * _Do improving model's capacity on chirality identification have positive impacts on the performance on benchmarks such as QM9?_
> > >
> > > We emphasize that ChIRo was designed to improve _chiral_ representation learning for _chirality-sensitive_ property prediction tasks, rather than for general molecular property prediction tasks. Notably, the regression targets in QM9 are invariant to tetrahedral chirality, since quantum mechanical energies for single, non-interacting molecules are invariant to the reflection transform. Thus, using ChIRo would not lead to any performance gains on QM9 in particular.  Also, QM9 energies are conformation-specific; since ChIRo is InterRoto-invariant, it would not be expected to perform well on these conformer-specific tasks.
> > >
> > > At the same time, because ChIRo can be easily packaged with any 2D GNN as an add-on, it will not hurt their performance on common 2D molecular property prediction benchmarks (e.g., MoleculeNet).

---

> > > > ### Comment · Reviewer_3jRF · 2021-12-07
> > > > **thanks for the clarification**
> > > >
> > > > Thanks for the clarification. And I guess we will see the results of such add-ons in the short future :)
> > > >
> > > > I have raised my scores to an apprant acceptance, sorry for the late.

---

> > > ### Comment · Reviewer_3jRF · 2021-12-07
> > > **still feel a bit strange, but it is a minor point**
> > >
> > > "It is also a form of contrastive learning in our current formulation, as positive pairs (conformers sharing the same chiral identity) are pulled together, whereas negative pairs (conformers with different chiral identities) are pushed apart in the latent space".
> > >
> > > Well, it seems to me that conformers are always different instances (they have unidentical chemical/bio properties), no matter whether they share the same chiral identity or not. It is somewhat similar to the "Generalization to an arbitrary number of positives" in section 3.2.2 of (Khosla et al., 2020), but I am still not fully convinced.

---

> > ### Comment · Reviewer_3jRF · 2021-12-07
> > **alright, I am a bit convinced**
> >
> > alright, I am a bit convinced.
> >
> > Though it seems that omitting the chiral atom tags is a less commonly used setting, as they are pretty easy to get using tools like RDKit. I believe the design of torsion encoder might shed light on the future work of geometrical modelling of molecules/proteins etc.

---

### Official Review · Reviewer_Viqv · 2021-11-03

**Correctness:** 3
**Technical Novelty And Significance:** 2
**Empirical Novelty And Significance:** 3
**Recommendation:** 5
**Confidence:** 4

**Main Review:**

Overall I liked the paper's idea. It is well written, easy to follow, and the provided list of related work was relevant. Even though technically it might be incremental, it might be able to provide an interesting insight and informative approach to see.

However, a big concern is: it is unclear which parts are technically novel, and actually brings any new flexibility because the paper combines several existing ideas. I understand providing a new effective combination of existing ideas is also an important contribution, but for this paper, I'm still not sure on this point of the significance of the provided combinations. The current form just seemed a minor straightforward modification by combining existing GNN design patterns to achieve SE(3) invariance. It is also unclear why other referred models such as SphereNet (Liu+, 2021), GemNet (Klicpera+ 2021), GeoMol (Ganea+ 2021) are insufficient for this targeted problem.

Technically speaking, the components are (1) 2D graph encoders, (2) bond distance and bond angle encoders, and (3) torsion encoders.

- (1) are a standard design pattern by EdgeConv and GAT layers
- (2) largely relies on the idea by Winter et al (2021)
- (3) largely relies on the idea by GeoMol (Ganea+ 2021) that considers torsional geometric handlings.

(1) are standard design patterns by EdgeConv + GAT layers, and (2) follows Winter et al (2021) to drop out any information on atom-atom distances and the like. To handle the chirality, the most important aspect would be torsion angles of (3), but comparisons to the original GeoMol (Ganea+ 2021) are presented only in "A.8 Additional Ablations" with a confusing statement of "It is not strictly necessary that ChIRo learn the weight coefficients or phase shifts in order to distinguish enantiomers".

So, the presented model seemed a minor modification of existing ideas to achieve SE(3)-invariance. The novelty is somewhat still unclear. Several natural questions are

- For example, can we include GeoMol (Ganea+ 2021) in Table 2?
- If (2) is from Winter et al (2021), and (3) is mainly from Ganea+ (2021), can we consider a direct combination of these two as a basline in all experiments? OR this work is exactly doing this...? If this won't work, why?
- Achieving SE(3)-invariance itself is the purpose? I understood SchNet and DimeNet will fail, but cited existing models such as SphereNet (Liu+, 2021), GemNet (Klicpera+ 2021), GeoMol (Ganea+ 2021) are also insufficient for some reasons? How they are different from the presented work?


*Remarks*: For ML practitioners who try to apply GNNs to life science problems or chemistry problems, the chirality of molecules is very important issue but also quite a big technical hurdle. When a molecule takes a 3D shape, we have a degree of freedom to rotate every C-C single bond. However, we have enantiomers, that is, non-superposable isomers by these operations but indistinguishable by 2D chemical formulas. These mirror-image molecules are very important both in life science (many drugs are enantiomers because interacting molecules in living cells also have chiral centers) and chemistry (to produce such drug compounds efficiently we need asymmetric synthesis producing only one of such enantiomers).


**Summary Of The Paper:**

This paper presents a novel SE(3)-invariant GNN model for predicting 3D geometry-dependent physicochemical properties of molecules. In particular, this method focuses on an important issue of how to handle the chirality of molecules in molecular GNNs. When a molecule takes a 3D shape, we have a degree of freedom to rotate every C-C single bond. To distinguish enantiomers (mirror-image molecule pairs), we also need to see whether two molecule pairs are superposable by these single-bond rotation operations. The developed method proposed a torsion-angle encoder having 1) an invariance to rotations about internal molecular bonds and 2) the ability to learn molecular chirality. Several empirical results are also reported.


**Summary Of The Review:**

The idea and the focused problem of the paper is very interesting and can be impactful. However, technically speaking, the current form seemed quite incremental, and a minor modification of existing ideas to achieve SE(3)-invariance.

---

> ### Author Response · Authors · 2021-11-16
> **Authors' Response to Reviewer Viqv (Part 1)**
>
> We thank the reviewer for their comments and suggestions. We have provided detailed responses to each of their specific comments below. We ask the reviewer to consider increasing their recommendation if we fully address their concerns.
>
> _However, a big concern is: it is unclear which parts are technically novel, and actually brings any new flexibility because the paper combines several existing ideas. I understand providing a new effective combination of existing ideas is also an important contribution, but for this paper, I'm still not sure on this point of the significance of the provided combinations. The current form just seemed a minor straightforward modification by combining existing GNN design patterns to achieve SE(3) invariance. It is also unclear why other referred models such as SphereNet (Liu+, 2021), GemNet (Klicpera+ 2021), GeoMol (Ganea+ 2021) are insufficient for this targeted problem._
>
> Our model achieves SE(3)-invariance less so through specific GNN design patterns, but more so through the choice of representing 3D molecular structures in terms of their internal coordinates (bond distances, bond angles, and torsions), which are natively E(3)- (distances and angles) or SE(3)-invariant (torsions). Many papers use molecular internal coordinates, and we do not claim novelty here. However, we _do_ claim novelty in processing the 3D internal coordinates, in particular the torsions, to achieve invariance to internal bond rotations (InterRoto-invariance) whilst still learning tetrahedral chirality from the 3D conformer structure. SphereNet (and GemNet) are natively SE(3)-invariant, but _not_ InterRoto-invariant. Thus, when learning tetrahedral chirality, SphereNet will get confused between the structural differences specifically caused by chirality, and the structural differences caused by simple bond rotations (e.g., conformational changes) which do not change the underlying chirality (Figure 3). This is illustrated by SphereNet’s 10% drop in accuracy in the l/d classification task when only training on one conformer per enantiomer (Table 2), as well as by our new experiments in A.10 (please see our response to Reviewer 1). We claim and demonstrate empirically that SE(3)-invariant models that are _not_ InterRoto-invariant will have limited ability to learn expressive and generalizable chiral representations of molecules, at least without extensive conformer enumeration and data augmentation.
>
> GeoMol (Ganea et al., 2021) is a conformer generator, _not_ a property prediction model. They proposed a scheme (Equation 8) to aggregate coupled torsions to avoid internal coordinate redundancies when generating conformers from the 2D graph. However, their weighted sums ( $\alpha^{cos}$ and $\alpha^{sin}$ in Equation 8) are _not_ InterRoto-invariant. Our first novel technical contribution is encoding the norm $||\alpha^{cos}, \alpha^{sin}||$ (e.g., the radius) to achieve InterRoto-invariance. Moreover, as we discuss in the subsections leading up to Equations 10-11 and prove in Appendix A.2, without the additions of Equations 10-11, this radius $||\alpha^{cos}, \alpha^{sin}||$ is invariant to chirality. Thus, our second novel technical contribution is to also learn phase shifts (Eqs. 10-11) for each coupled torsion in order to distinguish enantiomers and express tetrahedral chirality. Therefore, the two core contributions of ChIRo, namely its InterRoto-invariance and its expressivity toward chirality, are both our own novel contributions.

---

> > ### Author Response · Authors · 2021-11-16
> > **Authors' Response to Reviewer Viqv (Part 2)**
> >
> > _Technically speaking, the components are (1) 2D graph encoders, (2) bond distance and bond angle encoders, and (3) torsion encoders._
> >
> > * _(1) are a standard design pattern by EdgeConv and GAT layers_
> >
> > * _(2) largely relies on the idea by Winter et al (2021)_
> >
> > * _(3) largely relies on the idea by GeoMol (Ganea+ 2021) that considers torsional geometric handlings._
> >
> > _(1) are standard design patterns by EdgeConv + GAT layers_
> >
> > We have purposefully chosen to use standard, off-the-shelf 2D GNN encoders in order to emphasize that our torsion encoding scheme is the driving factor leading to ChIRo’s ability to express tetrahedral chirality and its InterRoto-invariance. Furthermore, this makes our model flexible with respect to the chosen 2D GNN: the node states can be embedded using any current or yet-to-be-developed graph encoder. This makes the core components of ChIRo modular and able to be added to existing 2D GNN predictors in order to make them expressive toward tetrahedral chirality.
> >
> > _(2) follows Winter et al (2021) to drop out any information on atom-atom distances and the like._
> >
> > We have included the bond distance and bond angle encoders, which as pointed out are only marginally changed from Winter et al. (2021), in order to provide just _one_ way for ChIRo to have greater access to the full geometric information in the conformer, _if ChIRo were to be used as-is in more mainstream molecular property prediction tasks._ This is not the _only_ way, or even the best way, for ChIRo to exploit bond distances and bond angles. There are a number of other GNNs, for instance Flam-Shepherd et al. (2020), which explicitly use bond distances and bond angles during message passing to improve general predictive performance. Exactly how ChIRo uses bond distances and angles is largely irrelevant for chiral property prediction, since bond distances and angles are E(3)-invariant and thus do not contribute to ChIRo’s ability to learn chirality. Since we have introduced ChIRo primarily as a way of learning robust representations of chirality, we are less concerned with how we process 2D and E(3)-invariant information. We consider this as a helpful _feature_, rather than a limitation, since it facilitates ChIRo’s integration into existing GNN architectures and workflows that use bond distances and angles in their own particular ways.
> >
> > _To handle the chirality, the most important aspect would be torsion angles of (3), but comparisons to the original GeoMol (Ganea+ 2021) are presented only in "A.8 Additional Ablations" with a confusing statement of "It is not strictly necessary that ChIRo learn the weight coefficients or phase shifts in order to distinguish enantiomers"._
> >
> > As expounded upon and clarified previously in our Part 1 response, GeoMol is a conformer generator and not a property prediction model. Directly adopting GeoMol’s coupled torsion encoding scheme would not achieve InterRoto-invariance, nor expressivity toward chirality, as mentioned earlier in our response. Appendix A.8 highlights a small difference between ChIRo’s default torsion aggregation scheme and that of GeoMol: whereas GeoMol used weight coefficients that are generated by a random, untrained network, we choose to learn the parameters of the network that predicts the weighting coefficients. This choice ultimately has little to no influence on ChIRo’s performance, as evidenced in A.8.
> >
> > However, it is critical that ChIRo learn the phase-shifts, rather than have these also predicted by an untrained network, which we also empirically demonstrate in A.8. Since GeoMol does not use phase-shifts (this is our contribution), a direct comparison here is infeasible.

---

> > > ### Author Response · Authors · 2021-11-16
> > > **Authors' Response to Reviewer Viqv (Part 3)**
> > >
> > > _So, the presented model seemed a minor modification of existing ideas to achieve SE(3)-invariance._
> > >
> > > One of our main contributions is a significant modification of GeoMol’s torsion encoding scheme to achieve _InterRoto-invariance_, which is a novel contribution and, as we demonstrate, critical for ChIRo’s strong chiral expressivity and superiority to SE(3)-invariant networks like SphereNet in chiral property prediction tasks (Table 2). SE(3)-invariance is guaranteed simply by our choice of using internal coordinate representations of the 3D geometry. Please note that InterRoto-invariance is entirely distinct from SE(3)-invariance.
> > >
> > > _The novelty is somewhat still unclear. Several natural questions are_
> > > * _For example, can we include GeoMol (Ganea+ 2021) in Table 2?_
> > >
> > > No: since GeoMol is a conformer generator, it is not directly applicable to chiral property prediction.
> > >
> > > * _If (2) is from Winter et al (2021), and (3) is mainly from Ganea+ (2021), can we consider a direct combination of these two as a basline in all experiments? OR this work is exactly doing this...? If this won't work, why?_
> > >
> > > Directly combining the torsion aggregation from Ganea et al. (2021) with Winter et al. (2021)’s encoders would yield an SE(3)-invariant, _non_-InterRoto invariant encoder that only slightly builds upon Winter et al. (2021)’s own torsion (dihedral) encoder. Namely, whereas Winter et al. (2021) sums the latent encodings of every torsion in the conformer, Ganea et al. (2021) aggregates coupled torsions with a weighted sum to define a new angle ($\alpha$) to describing a single, (non-physical) rotation angle of the common bond. While one could encode these $\alpha$ angles in a manner analogous to Winter et al. (2021)’s torsion encoder, this would yield an SE(3)-invariant encoder that is _not_ InterRoto-invariant, similar to SphereNet, which we have shown to have limited chiral expressivity through multiple empirical demonstrations (Table 2).
> > >
> > > It is true that ChIRo does combine elements of Winter et al. (2021) and Ganea et al. (2021), but we specifically introduce InterRoto-invariance and the ability to robustly express chirality through completely novel technical additions to these existing frameworks.
> > >
> > > _Achieving SE(3)-invariance itself is the purpose? I understood SchNet and DimeNet will fail, but cited existing models such as SphereNet (Liu+, 2021), GemNet (Klicpera+ 2021), GeoMol (Ganea+ 2021) are also insufficient for some reasons? How they are different from the presented work?_
> > >
> > > Achieving _InterRoto-invariance_ while remaining expressive toward chirality is the purpose of ChIRo (Figure 1B). SE(3)-invariance, which ChIRo also achieves, on its own is not sufficient for a 3D GNN to learn generalizable representations that achieve good empirical performance on supervised learning tasks that are sensitive to chirality. SchNet and DimeNet(++) are E(3)-invariant, and thus cannot express chirality due to being invariant to the reflection transform. They are also not InterRoto-invariant. SphereNet and GemNet are SE(3)-invariant, but are not InterRoto-invariant. GeoMol is not a property prediction model, and it is not InterRoto-invariant. Out of all the 3D neural network models that have the capacity to learn tetrahedral chirality (e.g., are not E(3)-invariant), _only_ ChIRo is both SE(3)-invariant and InterRoto-invariant.

---

> > ### Comment · Reviewer_Viqv · 2021-11-19
> > **Thanks for the detailed explanations**
> >
> > I understood that GeoMol (Ganea et al., 2021) is a conformer generator, and it is directly incomparable. As for my comment as
> >
> > > Technically speaking, the components are (1) 2D graph encoders, (2) bond distance and bond angle encoders, and (3) torsion encoders.
> > > (1) are a standard design pattern by EdgeConv and GAT layers
> > > (2) largely relies on the idea by Winter et al (2021)
> > > (3) largely relies on the idea by GeoMol (Ganea+ 2021) that considers torsional geometric handlings.
> >
> > so (1) and (2) (intentionally) followed a standard design pattern, and the point here is about (3) the torsion angle encoder of the ChIRo.
> >
> > I initially thought that rewriting Eq (8) of GeoMol (Ganea et al., 2021) into Eq (11) by incorporating the learned phase shifts of Eq (10) was the primary contribution from a technical viewpoint. But the paper mentioned "It is not strictly necessary that ChIRo learn the weight coefficients or phase shifts in order to distinguish enantiomers", so I was very confused. I thought that the torsion angle aggregation by Eq(8) or Eq(10) would be important feature for characterizing the chirality.
> >
> > But in fact, rather than Eq(10) and (11), **using the radius (using the norm of torsion angles) as Eq(12)** is the primary contribution? I understood that taking the norm brings the InterRoto-invariance.
> >
> > I will discuss these points with other reviewers and ACs.

---

> > > ### Author Response · Authors · 2021-11-22
> > > **Follow-up response to Review Viqv**
> > >
> > > > I initially thought that rewriting Eq (8) of GeoMol (Ganea et al., 2021) into Eq (11) by incorporating the learned phase shifts of Eq (10) was the primary contribution from a technical viewpoint. But the paper mentioned "It is not strictly necessary that ChIRo learn the weight coefficients or phase shifts in order to distinguish enantiomers", so I was very confused. I thought that the torsion angle aggregation by Eq(8) or Eq(10) would be important feature for characterizing the chirality.
> > >
> > > We have clarified this statement in our revised paper to better align with the empirical results shown in Table 16 of Appendix A.8. Namely, learning the phase shifts is critical for the superior performance of ChIRo. The statement now reads:
> > >
> > > “It is not strictly necessary that ChIRo *learn* the weight coefficients $c_{ij}^{(xy)}$ in order to distinguish enantiomers. ChIRo can still learn chiral representations and preserve invariance to internal bond rotations if each $c_{ij}^{(xy)}$ is generated by a network with random, untrained parameters. However, this is not the case for the phase shifts $\varphi_{ixyj}$, which must be learned for good performance.”
> > >
> > > > But in fact, rather than Eq(10) and (11), **using the radius (using the norm of torsion angles) as Eq(12)** is the primary contribution? I understood that taking the norm brings the InterRoto-invariance.
> > >
> > > Yes, using the norm in Eq. 12 brings in the InterRoto-invariance, and is **one** of the ***two*** primary technical contributions. The ***second*** primary technical contribution is the addition of phase shifts to each torsion in Eqs. 10-11, which allows the model to express chirality. Appendix A.2 proves that these phase shifts are necessary to distinguish enantiomers in our InterRoto-invariant framework.

---

### Official Review · Reviewer_pDCs · 2021-11-03

**Correctness:** 3
**Technical Novelty And Significance:** 3
**Empirical Novelty And Significance:** 3
**Recommendation:** 6
**Confidence:** 3

**Main Review:**

Overall the model sounds reasonable and the results on multiple tasks are promising. I have several minor questions about empirical studies.
- “ Although SE(3)-invariant 3D GNNs, such as the recently proposed SphereNet (Liu et al., 2021) and GemNet (Klicpera et al., 2021), can
in theory learn chirality, their expressivity in this setting has not been explored.” Give that both SphereNet and GemNet can learn chirality, why only SphereNet is compared in experiments? It is better to compare with both methods.
- “ We emphasize that we view this classification task as necessary but not suf￾ficient to demonstrate that a model can learn a meaning￾ful representation of chiral molecules.” Can you establish a bit more about this? Why not sufficient?
- SphereNet is comparable to Chiro in Table 1, but much worse in Table 2. Is this mainly because the differences between two tasks?  I’d like to see some analysis here. Why is Chiro much better in the second task? It would be great if you can show several case studies to support Chiro.

**Summary Of The Paper:**

This work focuses on representation learning for molecules with stereochemistry and designs an SE(3)-invariant model that processes torsion angles of a 3D molecular conformer. In particular, the authors explicitly model conformational flexibility by integrating a novel
type of invariance to rotations about internal molecular bonds into the architecture, mitigating the need for multi-conformer data augmentation.

**Summary Of The Review:**

Overall it is a nice work, but I’d like to see some deep analysis on experiments, e.g., why the proposed model is better, and in which cases, etc.

---

> ### Author Response · Authors · 2021-11-16
> **Authors' Response to Reviewer pDCs**
>
> We thank the reviewer for their comments and suggestions. We have provided detailed responses to each of their specific comments below. We ask the reviewer to consider increasing their recommendation if we fully address their concerns.
>
> _“Although SE(3)-invariant 3D GNNs, such as the recently proposed SphereNet (Liu et al., 2021) and GemNet (Klicpera et al., 2021), can in theory learn chirality, their expressivity in this setting has not been explored.” Give that both SphereNet and GemNet can learn chirality, why only SphereNet is compared in experiments? It is better to compare with both methods._
>
> The open-source code for GemNet (https://github.com/TUM-DAML/gemnet_pytorch) was released only very recently (and after the ICLR 2022 submission deadline). For this reason, we did not include GemNet in our comparisons. Since GemNet is also an SE(3)-invariant 3D graph neural network, we expect the performance in chiral property prediction to be similar to that of SphereNet, which we have treated as a prototypical SE(3)-invariant model.
>
> _“ We emphasize that we view this classification task as necessary but not sufficient to demonstrate that a model can learn a meaningful representation of chiral molecules.” Can you establish a bit more about this? Why not sufficient?_
>
> This comment refers to the toy classification task of tetrahedral chiral centers as rectus (R) versus sinister (S), according to Cahn-Ingold-Prelog rules of chemical identification. Importantly, the R/S designations of chiral molecules can be trivially evaluated with cheminformatics software (e.g., RDKit), and they only indicate the handedness of a tetrahedral chiral center. In fact, the R/S labels are sometimes included as atom chiral tags in node features, as discussed in our Related Work section. Testing a model’s ability to learn R/S distinctions serves as a simple litmus test for the basic ability of the model to distinguish enantiomers; one can view it as testing whether the model can learn the one-hot encoded chiral atom tags from the 3D structure. It does _not_ evaluate the model’s ability to learn how chirality influences the downstream chiral properties of the molecule, such as optical rotation or protein binding affinity. Hence, we only use the R/S task to show that the E(3)-invariant SchNet and DimeNet++ are fundamentally limited in their ability to learn chirality, whereas the SE(3)-invariant SphereNet and ChIRo do have some capacity to learn chirality.
>
> _SphereNet is comparable to Chiro in Table 1, but much worse in Table 2. Is this mainly because the differences between two tasks? I’d like to see some analysis here. Why is Chiro much better in the second task? It would be great if you can show several case studies to support Chiro._
>
> Yes, this gap in performance is directly attributable to the differences between the tasks. As discussed in our immediately preceding response, the R/S classification task is a toy litmus test of chiral perception only, whereas the tasks in Table 2 are designed to empirically compare the expressiveness, and hence generalizability, of these models when learning the biochemical effects of tetrahedral chirality. The R/S task could be trivially solved by RDKit, whereas the other tasks are significantly more difficult.
>
> We believe much of the drop in SphereNet’s performance in the latter tasks is attributable to SphereNet confusing differences in chiral identity with differences in conformational structure when learning tetrahedral chirality from 3D molecular structures. Since ChIRo is InterRoto-invariant, it is not as susceptible to this confusion. To support this hypothesis, we have added an appendix section (A.10) that quantifies how frequently SphereNet predicts the same label for each conformer of a given enantiomer in the test sets for the l/d classification task, compared to ChIRo. To summarize, SphereNet predicts the same label (regardless of this label’s accuracy) for each of the 5 conformers per enantiomer for only approximately 32% of the enantiomers across the test sets, compared to 93% for ChIRo. When SphereNet is trained on only 1 conformer per enantiomer, its labeling consistency drops to 14%, whereas ChIRo’s labeling consistency remains roughly the same at 92%. Furthermore, we have analyzed how SphereNet misclassifies enantiomers according to their sign of optical rotation when we manually rotate bonds near the chiral center, specifically analyzing three pairs of enantiomers in the test set (of fold 1) whose conformers were each originally classified correctly by SphereNet (e.g., before the manual bond rotations). We generate confusion matrices for each of these three pairs of enantiomers in A.10, which collectively show that SphereNet gets confused for approximately 10% - 20% of the conformers with rotated bonds, depending on the particular enantiomer.
>
> Note that SE(3)-invariant networks like SphereNet are not designed to be InterRoto-invariant; they are designed to be sensitive to dihedrals.

---

### Author Response · Authors · 2021-11-16
**Authors' Summary of Revisions**

The authors thank the reviewers for taking the time to provide detailed comments, to ask probing questions, and to suggest additional clarifying studies for our submission. Our primary responses to each specific issue raised by the reviewers are commented below each particular review.

Here, we summarize the major adjustments made to our submission to clarify our paper’s novel contributions and overall significance for representation learning on chiral molecules.

* In the introduction, we have clarified that “We do not consider common MoleculeNet (Wu et al., 2017) benchmarks, as our focus is on tasks where the effects of chirality are more distinguishable from experimental noise.” This serves to emphasize that ChIRo is specifically designed to improve the expressivity and generalizability of 2D GNNs when learning chirality-dependent functions. At the same time, ChIRo can be used as an add-on for any current SOTA or yet-to-be-developed 2D GNN. We clarify this point in the “2D Graph Encoder” section with our additional statement that “Any 2D GNN suffices to embed node states.”

* In the caption accompanying Figure 2, we have added clarifying comments to help the reader interpret the geometric visualization.

* In the “Torsion Encoder” section, we have made it explicitly clear what our novel technical contributions are compared to Ganea et al. (2021) (which is a conformer generator, not a property-prediction model). Namely, we modified their torsion aggregation scheme (Equations 8-9) to specifically introduce InterRoto-invariance directly into ChIRo’s architecture, as well as to enable ChIRo to learn and express tetrahedral chirality.

* In the caption accompanying Figure 3, we have clarified what is presented in the middle row “Original + Reflected Conformers”.

* In the experimental section “Classifying Tetrahedral Chiral Centers as R/S”, we have clarified that this is a toy task, given that 1) RDKit can already trivially classify chiral centers as R/S (which is not the case for the harder tasks in Table 2), and 2) that these R/S labels could themselves be added to the node features as chiral atom tags.

* In the “Ablation Study” section, we have clarified that we _expected_ the result that adding chiral tags to the node/bond features does not improve ChIRo’s performance. In particular, we expected this result given ChIRo's empirically demonstrated improvement over 2D GNNs with chiral tags (Table 2).

* Our most significant addition is in the new appendix section A.10, where we perform additional analysis and experiments with SphereNet on the l/d classification task to highlight that ChIRo’s quantitative performance improvement over SphereNet can be directly attributed to ChIRo’s InterRoto-invariance, which makes ChIRo consistently label different conformers of the same stereoisomer. In fact, for the enantiomers in the test sets of the l/d classification task, ChIRo labels all 5 conformers (per enantiomer) with consistent labels for 93% of the enantiomers across the 5 test folds, compared to only 32% for SphereNet. We further demonstrate that even for enantiomers which are consistently and accurately labeled by SphereNet, manually rotating bonds near the chiral carbon center causes SphereNet to mislabel approximately 10-20% of the rotated conformers.

---

> ### Comment · Reviewer_Viqv · 2021-11-19
> **A quick question on the first point**
>
> Thanks for the revision and answering the questions.
>
> I have a quick question on the first point to make things clear. So the ChIRo is essentially a 2D GNN even though it used bond distances, angles, torsion angles from the given 3D geometry, right?
>
> Then I felt that MoleculeNet (Wu et al., 2017) can be used because the most bioactivities would be chirality-dependent. So "when learning chirality-dependent functions" means much narrower targets than situations of MoleculeNet...? **Two chiral molecules have the exactly same chemical properties, except when reacting with other chiral compounds.** If GNNs take a molecule as an input without any information on the interacting pair, any representation cannot model any effect from the chirality, doesn't it...?
>
> *Remark:* Geometric GNNs such as SchNet, DimeNet++, GemNet, SphereNet are mostly for the task of approximating computational chemistry calculations for a given specific geometry such as QM9 or the like. In this case, the target properties are not InterRoto-invariant because the properties like energy change when the geometry changes. Moreover, we don't need to distinguish chiral molecules in these cases, because chiral molecules have the same properties in principle. I guess this is why these GNNs are not InterRoto-invariant.

---

> > ### Author Response · Authors · 2021-11-22
> > **Clarifications addressed to Reviewer Viqv**
> >
> > We hope the following responses can help clarify these questions.
> >
> > > So the ChIRo is essentially a 2D GNN even though it used bond distances, angles, torsion angles from the given 3D geometry, right?
> >
> > ChIRo is not a 2D GNN; ChIRo is a 3D GNN that is invariant to the conformational differences specifically caused by bond rotations. Although ChIRo uses a 2D GNN to embed the node states, it uses complex, learned aggregations of 3D internal coordinates (especially torsions) during the encoding and readout phases. Using bond distances, angles, and/or torsion angles from the 3D geometry is the underlying design principle behind most 3D GNNs.
> >
> > > Then I felt that MoleculeNet (Wu et al., 2017) can be used because the most bioactivities would be chirality-dependent. So "when learning chirality-dependent functions" means much narrower targets than situations of MoleculeNet...?
> >
> > It is true that bioactivities may be chirality-dependent. However, experimental measurements of bioactivity, like those in the MoleculeNet datasets, are often too noisy for the effects of chirality to be meaningfully learned. It is therefore unlikely that we would see significant differences in the models’ performances for these particular datasets, especially since **both** enantiomers of the pair (or multiple stereoisomers in general) are **not** typically included in the data. We point to [1], who observed these limitations in MoleculeNet’s lipophilicity dataset. We have amended our revision to clarify this point: “We do not consider common MoleculeNet (Wu et al., 2017) benchmarks, as our focus is on tasks where the effects of chirality are more distinguishable from experimental noise.”
> >
> > For these reasons, we specifically created our datasets and designed our tasks in a direct attempt to isolate the nuanced effects of chirality from noise. We point to Appendix A.7.2 and A.7.3, which highlight our data filtering and generation efforts for the l/d classification task and the ranking enantiomers by their docking scores task. We also note that for each of our supervised datasets, we always include pairs of enantiomers. For the contrastive learning task, we even include multiple (>2) stereoisomers for a significant number of molecules in the dataset (Figure 6, appendix A.7). These efforts make our datasets significantly more applicable for benchmarking the *chiral* expressivity and generalizability of graph neural networks.
> >
> > > If GNNs take a molecule as an input without any information on the interacting pair, any representation cannot model any effect from the chirality, doesn't it...?
> >
> > This is not the case, since the task itself should capture those chiral interactions. For instance, our docking score task does not explicitly encode the chiral protein pocket; the interactions with the external chiral environment are implicitly captured by the regression targets.
> >
> > [1] Message passing networks for molecules with tetrahedral chirality. arXiv preprint arXiv:2012.00094.

---

> > > ### Comment · Reviewer_Viqv · 2021-11-23
> > > **Thanks for the clarification!**
> > >
> > > I acknowledge that I have read and understood the point.
> > >
> > > I was confused that ChIRo is InterRoto-invariant in the same sense that 2D GNNs are so. But I understood that ChIRo is not a 2D GNN, and return a different result if input geometry differences are not from bond rotations, and testing over a docking situation given a target sounds reasonable. If we fix the target system (binding pocket), 3D chirality would play an important role. Also, bioactivities are more complex properties even though they are chirality dependent, and the author's explanations are quite reasonable. Thanks again.

---

### Decision · Program_Chairs · 2022-01-20

**Decision:**

Accept (Poster)

**Comment:**

This work presents ChIRo, a method that incorporates 3D torsion angles of a molecular conformer to specifically handle chirality. Specifically ChIRo uses trigonometric functions to encode the torsion angles, which are invariant to bond torsion but sensitive to chirality, thus capable of distinguishing between enantiomers.  Overall, although not groundbreaking, we found the idea presented in the paper to be sufficiently novel and its ability to handle chirality to be of significant practical values.